# Cytocompatibility, Antimicrobial and Antioxidant Activity of a Mucoadhesive Biopolymeric Hydrogel Embedding Selenium Nanoparticles Phytosynthesized by Sea Buckthorn Leaf Extract

**DOI:** 10.3390/ph17010023

**Published:** 2023-12-22

**Authors:** Naomi Tritean, Luminița Dimitriu, Ștefan-Ovidiu Dima, Rusăndica Stoica, Bogdan Trică, Marius Ghiurea, Ionuț Moraru, Anisoara Cimpean, Florin Oancea, Diana Constantinescu-Aruxandei

**Affiliations:** 1Bioresources, Polymers and Analysis Departments, National Institute for Research & Development in Chemistry and Petrochemistry—ICECHIM, Splaiul Independentei No. 202, Sector 6, 060021 Bucharest, Romania; naomi.tritean@icechim.ro (N.T.); luminita.dimitriu@icechim.ro (L.D.); ovidiu.dima@icechim.ro (Ș.-O.D.); rusandica.stoica@icechim.ro (R.S.); bogdan.trica@icechim.ro (B.T.); marius.ghiurea@icechim.ro (M.G.); 2Faculty of Biology, University of Bucharest, Splaiul Independentei 91-95, 050095 Bucharest, Romania; anisoara.cimpean@bio.unibuc.ro; 3Faculty of Biotechnologies, University of Agronomic Sciences and Veterinary Medicine of Bucharest, Mărăști Blv., No. 59, 011464 Bucharest, Romania; 4Faculty of Chemical Engineering and Biotechnology, National University of Science and Technology Politehnica Bucharest, Splaiul Independenței nr. 313, 060042 Bucharest, Romania; 5Laboratoarele Medica Srl, str. Frasinului nr. 11, 075100 Otopeni, Romania; ionutmoraru@pro-natura.ro

**Keywords:** fungal chitosan, bacterial nanocellulose, phytosynthesized selenium nanoparticles, sea buckthorn leaf extract, gingival dysbiotic biofilm

## Abstract

Phytosynthesized selenium nanoparticles (SeNPs) are less toxic than the inorganic salts of selenium and show high antioxidant and antibacterial activity. Chitosan prevents microbial biofilm formation and can also determine microbial biofilm dispersal. Never-dried bacterial nanocellulose (NDBNC) is an efficient carrier of bioactive compounds and a flexible nanofibrillar hydrophilic biopolymer. This study aimed to develop a selenium-enriched hydrogel nanoformulation (Se-HNF) based on NDBNC from kombucha fermentation and fungal chitosan with embedded biogenic SeNPs phytosynthesized by an aqueous extract of sea buckthorn leaves (SbLEx)—SeNPsSb—in order to both disperse gingival dysbiotic biofilm and prevent its development. We determined the total phenolic content and antioxidant activity of SbLEx. Liquid chromatography–mass spectrometry (LC-MS) and high-performance liquid chromatography (HPLC) were used for the identification of polyphenols from SbLEx. SeNPsSb were characterized by transmission electron microscopy–energy-dispersive X-ray spectroscopy (TEM-EDX), dynamic light scattering (DLS), zeta potential, Fourier transform infrared spectroscopy (FTIR) and X-ray diffraction (XRD) in small- and wide-angle X-ray scattering (SAXS and WAXS). The hydrogel nanoformulation with embedded SeNPsSb was characterized by SEM, FTIR, XRD, rheology, mucin binding efficiency, contact angle and interfacial tension measurements. We also assessed the in vitro biocompatibility, antioxidant activity and antimicrobial and antibiofilm potential of SeNPsSb and Se-HNF. TEM, DLS and SAXS evidenced polydisperse SeNPsSb, whereas FTIR highlighted a heterogeneous biocorona with various biocompounds. The contact angle on the polar surface was smaller (52.82 ± 1.23°) than that obtained on the non-polar surface (73.85 ± 0.39°). The interfacial tension was 97.6 ± 0.47 mN/m. The mucin binding efficiency of Se-HNF decreased as the amount of hydrogel decreased, and the SEM analysis showed a relatively compact structure upon mucin contact. FTIR and XRD analyses of Se-HNF evidenced an interaction between BNC and CS through characteristic peak shifting, and the rheological measurements highlighted a pseudoplastic behavior, 0.186 N adhesion force and 0.386 adhesion energy. The results showed a high degree of cytocompatibility and the significant antioxidant and antimicrobial efficiency of SeNPsSb and Se-HNF.

## 1. Introduction

Resident bacterial species form a well-defined ecosystem in close symbiosis with the host. About 700 bacterial species are estimated to reside in the oral cavity [1]. Adherent (sessile) bacteria may show different resistance to antibiotics or to the defense mechanisms of the host organism compared to free-living (planktonic) bacteria of the same species. Dysbiotic bacterial biofilm (plaque) can induce gingivitis and periodontitis, which are pathological inflammatory responses of the gum tissues [2]. The abundant dysbiotic bacterial biofilms’ development on gingival tissue triggers innate immunity activation by microbial-associated molecular patterns (MAMPs)—lipopolysaccharides (LPS), antigens and virulence factors [3]. The failure of inflammation resolution leads to the release of damage-associated molecular patterns (DAMPs), activation of hyperinflammation responses and proliferation of pathogenic biofilms/dysbiotic biofilms [4]. Genetic risk factors, epigenetic effects and behavioral/environmental risk factors are involved in gingivitis evolution toward periodontitis [5]. There is practically no treatment to definitively cure these pathological conditions. Dysbiotic biofilm removal, resolution of inflammation (considering also the control of behavioral/environmental risk factors) and further restoration of health-promoting (symbiosis) biofilms are logical interventions derived from these largely accepted pathological mechanisms.

Plants are a rich source of bioactive compounds with various biological activities [6]. Sea buckthorn (*Elaeagnus rhamnoides*, syn. *Hippophae rhamnoides*) is a medicinal plant belonging to the Elaeagnaceae family. Different extracts from sea buckhorn’s leaves and fruits have demonstrated its antioxidant, antibacterial, anti-inflammatory and antiviral activities, as well as its potential for supporting tissue regeneration [7,8,9]. Therefore, natural plant extracts are a valuable alternative to classical antimicrobial treatments due to their low risk of side effects and various biological activities, which could potentiate the antimicrobial activity.

In addition to natural plant extracts, diverse nanostructures determine the resolution of dysbiotic biofilms and inflammation. Selenium (Se) is present in different chemical forms, organic and inorganic, which determine its bioavailability, functionality and toxicity [10]. It has been reported that the difference between the beneficial and toxic dose of selenium is very narrow [11]; therefore, new formulations are needed in order to reduce the risk of negative effects and to optimize the beneficial ones. Phytosynthesized selenium nanoparticles (SeNPs) were proven to prevent the formation of bacterial biofilm and to promote inflammation resolution especially due to their high antioxidant activity and also to the unique properties that result from their biosynthesis process [12,13,14,15,16,17].

Nanocellulose is a nanostructured biopolymer which has proven its effectiveness in different formulations intended for biomedical applications. It results from the mechanical processing of cellulose, which is the most abundant polysaccharide in nature and consists of long linear chains of β-D-glucose. A valuable alternative for plant cellulose, both from an environmental and economic point of view, is the production of bacterial cellulose (BC) by microbial processes. BC possesses high tensile strength, increased crystallinity and a high degree of polymerization, as well as high purity, with a water retention capacity which is more than 100 times higher in comparison with plant cellulose [18]. Moreover, never-dried bacterial nanocellulose (NDBNC) is an efficient carrier of different bioactive compounds that possess high antimicrobial, antioxidant and anti-inflammatory activities [19,20,21]. Another biopolymer intended for biomedical applications is chitosan, a linear polysaccharide derived from chitin, which composes the exoskeleton of crustaceans and the cell wall of fungi. Its physicochemical properties depend on the molecular mass and the degree of acetylation. Chitosan prevents bacterial biofilm formation and it can determine bacterial biofilm dispersal [22,23,24,25].

Mucoadhesive systems for the targeted delivery of drugs or for dermato-cosmetic treatments are intended to prolong the dwell time of the active substance at the application site and to facilitate interfacial contact and mass transfer between the bioactive mucoadhesive system and the absorbent surface, thereby improving their therapeutic performance. The formulations intended to inhibit dysbiotic biofilms should be nano or microformulated for an improved mass transfer and also to exhibit good mucoadhesive properties, as well as high flexibility and mechanical strength. Both chitosan and cellulose have mucoadhesive properties by forming hydrogen bonds with the glycosidic groups of mucin. Therefore, the chitosan–NDBNC formulations have become of great interest in regenerative medicine due to their diverse physicochemical and mechanical properties, as well as high biological performance [26].

The aim of this study was to develop a mucoadhesive hydrogel nanoformulation (Se-HNF) based on NDBNC from kombucha fermentation and fungal chitosan (CS) enriched with biogenic SeNPs phytosynthesized by an aqueous extract of sea buckthorn leaves (SeNPsSb), with high in vitro biocompatibility and antioxidant, antimicrobial and antibiofilm activities. To the best of our knowledge, this is the first study on SeNPs phytosynthesized by an aqueous extract of sea buckthorn leaves and on this type of formulation.

## 2. Results

### 2.1. Phenolic Profile of SbLEx

The phenolic profile of the aqueous extract of sea buckthorn leaves (SbLEx) was evaluated by the determination of the total polyphenols (TPC), flavonoids (TFC), hydroxycinnamic acids (HAT), anthocyanidins (TAC) and by LC-TOF/MS and HPLC-DAD analysis. The antioxidant activity of SbLEx was investigated by DPPH, FRAP and CUPRAC methods. These results are summarized in Table 1, Table 2 and Table 3. Of the total polyphenol content, about one third is represented by flavonoids, and about two thirds are represented by hydroxycinnamic acids. The antioxidant activity of SbLEx varies depending on the method used (Table 1).

**Table 1 pharmaceuticals-17-00023-t001:** Phenolic content and antioxidant activity of SbLEx ± standard error (SE).

	Sea Buckthorn Leaf Aqueous Extract (SbLEx)					
TPC (mg GAE */g Dry Weight)	TFC (mg QE */g Dry Weight)	HAT (mg ChAE */g Dry Weight)	TAC (mg C3GE */g Dry Weight)	DPPH (mM TE */g Dry Weight)	FRAP (mM TE/g Dry Weight)	CUPRAC (mM TE/g Dry Weight)
46.59 ± 0.93	4.23 ± 0.09	11.53 ± 0.23	0.0048 ± 0.0002	14.75 ± 0.20	258.2 ± 1.96	1196 ± 7.01

* GAE—gallic acid equivalents, QE—quercetin equivalents, ChAE—chlorogenic acid equivalents, C3GE—cyanidin-3-glucoside equivalents, TE—Trolox equivalents.

The LC-TOF/MS analysis identified a number of compounds including several polyphenols, the γ-Aminobutyric acid (GABA), an aldehyde (cirsium aldehyde) and a vitamin (thiamine) (Table 2). The LC-TOF/MS chromatograms and spectra of SbLEx are presented in Appendix A.

**Table 2 pharmaceuticals-17-00023-t002:** LC-TOF/MS analysis of SbLEx and quantification of catechin (C), epicatechin (EC) and quercetin 3-rutinoside (Que-rut) in SbLEx by HPLC analysis.

[M + H]^+^ *m/z*	Compound	mg Polyphenols × kg^−1^ of Dry Weight ± Standard Deviation (SD)
104.1	γ-Aminobutyric acid	
193.9	Quinic acid	
235.0	Cirsium aldehyde	
266.1	Thiamine	
291.2	Catechin (C)/Epicatechin (EC)	247.52 ± 7.38 for C/35.60 ± 0.920 for EC
449.3	Quercitrin/Quercetin-7-rhamnoside/Astragalin	
611.3	Quercetin 3-rutinoside (Que-rut)	337.06 ± 9.20 for Que-rut
625.2	Isorhamnetin-3-O-rutinoside/Isorhamnetin-3-O-glucoside/Isorhamnetin-3-O-glucoside-7-O-rhamnoside	
899.3	Prodelphinidin trimer	
949.3	Quercetin 3-O-(6-O-feruloylglucoside)-glucoside-7-O-rhamnoside	

Three flavonoids were quantified by HPLC-DAD analysis. Identification of these compounds was based on the retention time determined in the samples in comparison to the reference materials. The plots of the peak areas of the analytes versus the concentrations were found to be linear (Appendix A). Appendix A show the chromatogram of the catechin (C), epicatechin (EC) and quercetin 3-rutinoside (Que-rut) detected in SbLEx. The quantification of C, EC and Que-rut in SbLEx was based on an external calibration method, and the results are presented in Table 2. Que-rut and C were detected in much higher amounts than EC.

### 2.2. Physicochemical Characterisation of SbLEx and SeNPsSb

The conversion yield of selenite into SeNPsSb that was recovered by ultracentrifugation and determined by the method mentioned in Section 4.2.1. was 73.2 ± 1.8%. The phytosynthesis of SeNPsSb was investigated by the FTIR spectroscopy of SeNPsSb and the spectral comparison with the raw materials, respectively, sodium selenite Na_2_SeO_3_, ascorbic acid and SbLEx (Figure 1a). In the diagnostic region 4000–2600 cm^−1^, the SeNPsSb spectrum is characterized by consistent hydrogen bonds of hydroxyl and amino moieties with an absorption band at 3279 ± 280 cm^−1^. Hydrogen bonds could be formed between the non-bonding electrons of Se^0^ and hydrogen donors in biocorona [27,28] and contribute, together with van der Waals interactions, to the stabilization of the NPs. The C–H bond vibrations of alkyl chains, with the specific bands at 2918 and 2849 cm^−1^ [29,30,31], are better separated and sharper in SeNPsSb than in SbLEx, suggesting the presence of aliphatic chains like in long-chain amino acids and fatty acids, which are somewhat similar but less intense than those found in our previous SeNPs obtained by SCOBY/kombucha fermentation [27]. The SbLEx has a more intense band in the hydrogen bonds region compared with SeNPsSb, mainly due to interstitial water. Ascorbic acid was evident in this region in three main bands at 3522, 3410 and 3310 cm^−1^, characteristic for the three types of hydroxyl groups in the molecule, and, in the C–H region, four main bands at 2994, 2911, 2862 and 2716 cm^−1^ are present, for the symmetric and asymmetric vibrations of C–H in CH_3_ and CH_2_. Sodium selenite shows in this region only a small band at 3744 cm^−1^, assigned to hydrogen bonds in crystallization water. In the fingerprint region, Na_2_SeO_3_ shows a few strong absorption bands at 880, 839, 785, 714, 619 and 444 cm^−1^, specific for the selenite ion and individual Se=O and Se–O vibrations [27]. On the contrary, the SeNPsSb show small bands at 868, 766 and 719 cm^−1^ in the same region, shifted from the bands of Na_2_SeO_3_ towards the ones of ascorbic acid and SbLEx extract, suggesting the reduction and interaction of selenite with ascorbic acid and SbLEx and eventually some newly formed Se–O, Se–N and/or Se–C bonds [27,32,33]. Moreover, the main bands of SeNPsSb in the fingerprint region are similar to the carboxylic, amide and carbohydrate bands of SbLEx, but with wavenumbers shifted, suggesting contributions from proteins and saccharides to the biocorona which cause a few additional small bands such as that at 1788 cm^−1^.

The biocorona of SeNPsSb was studied by spectroscopic comparison with thiamine as a representative compound from the vitamins class and identified by LC-TOF/MS in SbLEx, as well as catechin as a flavonoid representant, BSA as a protein representant and palmitic acid from the fatty acids class (Figure 1b). The FTIR spectrum of SeNPsSb showed similarities to all four representative compounds. The similitude of the SeNPsSb spectrum with the thiamine spectrum is visible at the amine band around 1600 cm^−1^ and alcohol band around 1030 cm^−1^, while catechin resembles SeNPsSb at the polyphenol bands around 1612 and 1026 cm^−1^. The FTIR spectrum of SeNPsSb shows similarities with BSA as well, through the characteristic bands of amide I and II around 1640 and 1530 cm^−1^, respectively, and also the C–N band around 1070 cm^−1^. The C–H bands at 2918 and 2849 cm^−1^ in SeNPsSb are very similar with the bands at 2913 and 2845 cm^−1^ of palmitic acid, suggesting, therefore, the presence of aliphatic acids in the SeNPsSb biocorona.

The XRD diffractograms of SeNPsSb, performed at wide angle (WAXS) and small angle (SAXS), are presented in Figure 1c,d. In Figure 1c, the diffractogram of SeNPsSb is compared with the initial raw materials, respectively, the sea buckthorn leaves aqueous extract (SbLEx), sodium selenite (Na_2_SeO_3_) and ascorbic acid. In the WAXS mode, the SeNPsSb show a predominant amorphous character induced by the SbLEx extract, with the amorphous peak centered around 25.70° and a crystallinity index of 36%. The amorphous peak of SeNPsSb is displaced in comparison with the 20° peak in SbLEx, suggesting the interaction of the extract biocompounds with the reduced Se forming the structure known as biocorona. The diffractogram of SeNPsSb shows three more peaks at 19.18°, 21.48° and 23.30°, which are different than the crystalline peaks of Na_2_SeO_3_ or ascorbic acid, confirming the transformation of the raw materials. The crystallite size distribution, automatically determined by the PDXL software with the Scherrer equation, is presented at the bottom of Figure 1c and it evidenced four populations of crystallites with crystallite sizes as multiples of about 3 nm, respectively, 3.5 nm, 6.7 nm, 8.9 nm and 12.3 nm, suggesting that the SeNPs’ grains are built as clusters of 3 nm crystallites. The SAXS diffractograms presented in Figure 1d were performed at three different angle ranges and showed different small-angle diffraction peaks between 0.164° and 0.956° convoluted in a signal similar to other types of nanoparticles [34], whereas the polydispersity of nanoparticles with diameters between 6 and 100 nm is presented in Figure 1e, with two main populations centered around 17 nm and 24 nm and two secondary populations centered around 10 nm and 50 nm. The SAXS particle sizes appear to also be multiples of about 3 nm, respectively, around 9 nm, 12 nm, 18 nm, 24 nm, 27 nm and 54 nm, suggesting that, similar to WAXS, the SeNPs nanograins are formed by clusters of 3 nm crystallites.

TEM analysis highlighted quasi-spherical nanoparticles (Figure 2a) with sizes between 10 and 100 nm, considering the aggregates (Appendix A) surrounded by a biocorona as well, and the EDX analysis revealed the presence of C, O and Se as the main elements (Figure 2b). The weight and atomic percentages of each element are presented in Appendix A.

SeNPsSb were further investigated by DLS analysis using three independent models (cumulants, Pade–Laplace and SBL). The autocorrelation function and the fit by the corresponding model are presented in Appendix A. Using the cumulants model, we obtained a Z-average of 347.72 nm, with a polydispersity index (PDI) of 0.34 (a.u.), which highlights a polydisperse distribution, but in this case, the model did not fit well because the residues were out of limit. The next model—Pade–Laplace—detected two populations of 161.46 nm and 890.39 nm, the first population being predominant. The results of the SBL model analysis are summarized in Table 3 and were considered to be the most relevant to our system based on the autocorrelation function and TEM analysis. The distribution of the SeNPsSb mean diameter depending on intensity, volume and number are presented in Appendix A.

**Table 3 pharmaceuticals-17-00023-t003:** DLS analysis of SeNPsSb.

Mean Diameter (nm)	Intensity (%)	Volume (%)	Number (%)
4.28	-	-	100
104.81	-	97.32	-
142.71	24.98	-	-
514.29	-	2.68	-
657.6	75.02	-	-

The electronegative Zeta potential of −29.98 ± 0.421 mV suggested the high stability of SeNPsSb in aqueous solution (Appendix A), as electronegative and electropositive Zeta potentials lower than −30 and higher than +30 mV, respectively, are considered an index of high stability.

### 2.3. Biological Performance of SeNPsSb and Se-HNF

In order to investigate the biological performance of SeNPsSb and of the hydrogel with SeNPsSb (Se-HNF), we assessed the cytocompatibility and antioxidant activity, as well as the antimicrobial and antibiofilm potential.

Three different methods (DPPH, FRAP, CUPRAC) and three concentrations of SeNPsSb, i.e., 100, 200, 500 µg/mL, were used to determine the antioxidant activity (AOA) of SeNPsSb. AOA increased as the concentration of SeNPsSb increased (Figure 3). Using the DPPH method, the antioxidant activity increased from 105.9 ± 1.9 µM Trolox equivalents (TE) at the lowest concentration of SeNPsSb to 156.8 ± 2 µM TE at the highest concentration tested. In the case of the FRAP assay, the antioxidant activity at 100 µg/mL SeNPsSb was 113.1 ± 5 µM TE, reaching 266 ± 5 µM TE at the highest concentration of 500 µg/mL SeNPsSb. Using the CUPRAC method, the antioxidant activity of SeNPsSb increased from 305.5 ± 8.3 µM TE at 100 µg/mL SeNPsSb to 484.9 ± 16.53 at the highest concentration of 500 µg/mL SeNPsSb. All the increases were statistically significant.

Following the treatment of gingival fibroblasts with SeNPsSb concentrations between 0.1 and 10 µg/mL, a high degree of cytocompatibility was observed at relatively low concentrations (92.85 ± 2.67% for 0.1 µg/mL, 97.14 ± 0.66% for 0.5 µg/mL, 91.53 ± 3.47 µg/mL for 2.5 µg/mL SeNPsSb), but a cytotoxic effect was noted at the highest concentration tested (74.22 ± 1.4% for 10 µg/mL SeNPsSb) (Figure 4a). The CCK-8 assay results can be correlated with the fluorescence microscopy images acquired after performing the LIVE/DEAD test (Appendix A).

In order to determine the in vitro antioxidant activity, the highest concentrations of SeNPsSb found to be biocompatible were chosen, i.e., 0.5 and 2.5 µg/mL. Following treatment with 0.5 µg/mL SeNPsSb in the presence of the reactive oxygen species (ROS)-inducing agent, a decrease in ROS production to the level of the negative control (C−, untreated cells) was observed. By increasing the concentration of SeNPsSb to 2.5 µg/mL, a further decrease in ROS production to 81.9 ± 1.2 (% of C−) was noticed (Figure 4b). The results obtained by labeling and quantifying the total ROS with H_2_DCFDA can be correlated with the fluorescence microscopy images (Appendix A).

To investigate the cytocompatibility of the hydrogels based on the BNC-CS matrix enriched with 0.5 µg/mL and 2.5 µg/mL SeNPsSb (0.5 Se-HNF and 2.5 Se-HNF, respectively), we tested multiple hydrogel concentrations (10, 25, 50, 100, 500, 1000 µg/mL) prepared in complete culture medium. The BNC-CS matrix without SeNPsSb (HNF) was also tested in order to determine if there is a potential complementarity.

Following the determination of cell viability, a high degree of cytocompatibility of HNF was observed, with an increase in the number of viable metabolically active cells at 25 µg/mL compared with the control (107 ± 0.6% of C−). The same pattern was observed after treatment with 0.5 Se-HNF, with the cell viability reaching 106.4 ± 0.2% of C− at 25 µg/mL. In the case of the 2.5 Se-HNF treatment, all the concentrations were biocompatible, with only a slight decrease in cell viability down to 93.8 + 2.7% of C− at 1000 µg/mL (Figure 4c). The results obtained in Figure 4a can be correlated with the fluorescence microscopy images from Appendix A obtained after performing the LIVE/DEAD assay.

In order to investigate the in vitro antioxidant activity of HNF, 0.5 Se-HNF and 2.5 Se-HNF, we chose two hydrogel concentrations, respectively, 25 µg/mL, which was found to have the highest potential for stimulating cell proliferation, and the highest concentration, i.e., 1000 µg/mL (Figure 4), of HNF without SeNPsSb that did not decrease the ROS production. There was a slight increase in the amount of ROS at 1000 µg/mL HNF (148.3 ± 1.67% of C−) compared to C+ (140.6 ± 2.28% of C−), but it was not statistically significant. Following the treatment with 0.5 Se-HNF, there was a decrease in the ROS production to 123.3 ± 2% of C− at 25 µg/mL 0.5 Se-HNF and 123.6 ± 2.42% of C− at 1000 µg/mL 0.5 Se-HNF. Upon treatment with 2.5 Se-HNF, the amount of ROS reached 71.32 ± 1.43% of C− at 25 µg/mL hydrogel and 86.06 ± 1.32% of C− at 1000 µg/mL 2.5 Se-HNF (Figure 4d). The results can be correlated with the qualitative screening (Appendix A).

The aspect of the actin filaments (green fluorescence) and nuclei (blue fluorescence) indicates that the concentration of 2.5 µg/mL SeNPsSb did not induce changes in cell morphology compared to the negative cytotoxicity control (untreated cells), with the cytoskeleton being highly organized in a fibrillar arrangement. Also, no changes were observed in the morphology of gingival fibroblasts following the treatment with 25 and 1000 µg/mL HNF and 2.5 Se-HNF (Figure 5a–g).

The semiquantitative screening of the antimicrobial activity of SeNPsSb was determined using several Gram-positive bacterial strains (*B. cereus*, *E. faecalis*, *S. aureus*) and a yeast (*C. albicans*) (Appendix A). Six concentrations of SeNPsSb were tested, respectively, 0.05, 0.1, 0.2, 0.5, 1 and 2 mg/mL. The average diameter of the inhibition zone is presented in Table 4. At the lowest concentrations tested, i.e., 0.05, 0.1 and 0.2 mg/mL, *B. cereus* showed resistance to the action of SeNPsSb. Starting from 0.5 mg/mL, *B. cereus* was completely inhibited, with no significant differences in terms of the halo diameter when applying SeNPsSb concentrations higher than 0.5 mg/mL. In the case of *E. faecalis*, the concentration of 0.05 mg/mL SeNPsSb had no effect, but from 0.1 mg/mL to 0.5 mg/mL, an inhibition effect correlated with the applied SeNPsSb dose was observed: 0.390 ± 0.001 cm, 0.510 ± 0.006 and 0.86 ± 0.009 cm at 0.1 mg/mL, 0.2 mg/mL and 0.5 mg/mL SeNPsSb, respectively. The highest concentrations of SeNPsSb, respectively, 1 and 2 mg/mL, did not produce significant further increases in the inhibition zone for *E. faecalis*. *S. aureus* was inhibited at all tested concentrations, with the diameter of the halo between 0.86 ± 0.01 and 1.02 ± 0.06 cm and no statistically significant differences regarding the diameter of the inhibition zone following the application of different concentrations of SeNPsSb. *C. albicans* was resistant to all the tested concentrations of SeNPsSb.

By testing the effect of ten concentrations of SeNPsSb for each Gram-positive bacteria using the serial microdilution method, the minimum inhibitory concentration (MIC) was determined, as well as the IC50 value. Also, after determination of MIC, we assessed the minimum concentration for the eradication of the bacterial biofilm (MCEB). The MIC, IC50 and MCEB values (mg/mL) are presented in Table 5 and the IC50 plots are presented in Appendix A. *B. cereus*, *S. aureus* and *C. albicans* did not form biofilm, so the MCEB could only be determined in the case of *E. faecalis*.

Figure 5 shows the bacterial inhibition (% of the positive control, C+) in the case of all the tested concentrations of SeNPsSb.

For *B. cereus*, the MIC was the highest tested concentration, i.e., 1.5 mg/mL, and the IC50 value was 0.708 ± 0.031 mg/mL (Table 5). At SeNPsSb concentrations below the MIC, the inhibition decreased as the SeNPsSb concentration decreased, reaching 8.6 ± 0.6% at 0.05 mg/mL (Figure 6a).

In the case of *E. faecalis*, the MIC value was 1 mg/mL, and the IC50 was reached at a much lower concentration of SeNPsSb compared to *B. cereus*, i.e., 0.039 ± 0.008 mg/mL (Table 5). In this case, an interesting aspect is the fact that following the decrease in SeNPsSb concentration, the inhibition percentage decreased quite slowly in the concentration ranges 2–0.35 mg/mL, 0.2–0.05 mg/mL and 0.025–0.005 mg/mL, with the inhibition being still a significant 39.7 ± 1.3% at 0.005 mg/mL SeNPsSb (Figure 6b).

In the case of *S. aureus*, the MIC was 1 mg/mL (Table 6). The IC50 value of 0.288 ± 0.004 mg/mL (Table 5) was determined using the concentration range of 2–0.1 mg/mL, observing a dual effect of SeNPsSb, respectively, bacterial inhibition both at high and at low concentrations. From 0.05 mg/mL to 0.005 mg/mL, the inhibition percentage starts to increase, reaching 50.23 ± 1.5% at 0.01 mg/mL and 56.17 ± 1.8 mg/mL at 0.005 mg/mL (Figure 6c).

In the case of *C. albicans*, the MIC value was 0.75 mg/mL, while the IC50 value was 0.143 ± 0.001 mg/mL (Table 5). Also, the inhibition of microbial growth (% of C−) decreases as the concentration of SeNPsSb decreases (Figure 6d).

The MCEB was 0.1 mg/mL for *E. faecalis* (Table 5). The inhibition percentages of the *E. faecalis* biofilm at values below MCEB are presented in Table 5.

For the quantitative screening of the antimicrobial activity of the hydrogel formulations, the concentration of 25 µg/mL hydrogel was chosen, as it showed both a high degree of cytocompatibility and the highest potential for reducing ROS, and we varied the microbial cell density (between 1.5 × 10^7^ and 1.5 × 10^1^) in order to investigate the microbial growth prevention potential of the hydrogel formulations (Figure 7). The concentration of 1.5 × 10^0^ microbial cells served as negative control. An increase in the inhibition of the microbial growth (% of C+) with the decrease in the microbial cell density was observed 24 h after the treatment, mainly in the case of *B. cereus*, *E. faecalis* and *C. albicans*.

*B. cereus* grew at bacteria inoculations between 1.5 × 10^7^ and 1.5 × 10^3^ microbial cell density; at lower inoculations, no bacterial growth was recorded. The microbial growth inhibition by HNF started from 3.4 ± 1.05% at 1.5 × 10^7^ microbial cells, reaching 29.9 ± 3.7% at 1.5 × 10^3^ microbial cells. In the case of 0.5 Se-HNF, the inhibition of microbial growth increased from 11.9 ± 0.9% at 1.5 × 10^7^ microbial cells to 36.7 ± 6.5% at 1.5 × 10^3^ microbial cells. The most efficient nanoformulation for inhibition of *B. cereus* growth was 2.5 Se-HNF, which succeeded to inhibit 13.9 ± 1.7% of microbial growth at 1.5 × 10^7^ microbial cells and 41.3 ± 2% at the lowest microbial density tested of 1.5 × 10^3^ (Figure 7a).

*E. faecalis* grew at bacteria inoculations between 1.5 × 10^7^ and 1.5 × 10^1^ microbial cell density. HNF exhibited an inhibition of *E. faecalis* growth starting from 8.6 ± 1.1% at 1.5 × 10^7^ microbial cells and reaching 56.1 ± 2.3% at 1.5 × 10^1^ microbial cells. Following the 0.5 Se-HNF treatment, there was a higher microbial growth inhibition than HNF, from 12.6 ± 1.1% at 1.5 × 10^7^ to 59.63 ± 0.1% at 1.5 × 10^1^ microbial cells. The 2.5 Se-HNF treatment presented the highest potential for *E. faecalis* inhibition, starting from 21.4 ± 2.1% at 1.5 × 10^7^ microbial cells and reaching 59.3 ± 0.5% at 1.5 × 10^1^ microbial cells (Figure 7b).

In the case of both *B. cereus* and *E. faecalis*, the most significant differences between the tested formulations, especially between 2.5 Se-HNF and HNF, were observed at the highest microbial densities inoculated.

*S. aureus* grew at bacteria inoculations between 1.5 × 10^7^ and 1.5 × 10^3^ microbial cell density. The inhibition of *S. aureus* growth by HNF was 30.7 ± 2.3% at 1.5 × 10^7^ inoculation and increased up to 51.2 ± 1.6% at 1.5 × 10^3^ inoculation. The treatment with 0.5 Se-HNF inhibited the microbial growth by 45.1 ± 5% at 1.5 × 10^7^ microbial cells with a further increase up to 73.1 ± 0.5% inhibition at 1.5 × 10^3^ microbial cells. Following the 2.5 Se-HNF treatment at 1.5 × 10^7^ microbial cells, a marginally significant inhibition up to 56.6 ± 3% was observed compared to the 0.5 Se-HNF treatment. At the lowest microbial density, i.e., 1.5 × 10^3^ microbial cells, the inhibition by 2.5 Se-HNF reached 75.8 ± 0.3%, the difference being statistically significant only by comparison with HNF (Figure 7c).

*C. albicans* grew at microbial inoculations between 1.5 × 10^7^ and 1.5 × 10^5^ microbial cell density. At 1.5 × 10^7^ microbial cells, HNF inhibited the microbial growth by 0.06 ± 0.09%, reaching 24.5 ± 0.07% at 1.5 × 10^5^ microbial cells. A statistically significant increase in the inhibition of *C. albicans* growth in comparison with HNF treatment was observed in the case of 0.5 Se-HNF, which started from 4.1 ± 0.3% at 1.5 × 10^7^ microbial cells and reached 43.6 ± 1.1% at 1.5 × 10^5^ microbial cells. A further statistically significant increase in the microbial growth inhibition compared to 0.5 Se-HNF was upon 2.5 Se-HNF treatment, starting from 11.1 ± 0.3% at 1.5 × 10^7^ microbial cells and reaching 64.6 ± 2.0% at 1.5 × 10^5^ microbial cells (Figure 7d).

Due to the fact that the concentration of 2.5 µg/mL SeNPsSb was the highest tested concentration that proved to be cytocompatible and also showed the highest potential to reduce both ROS production and microbial growth by assessing the biological performance of Se-HNF in solution (Figure 4 and Figure 7), we further characterized the 2.5 Se-HNF hydrogel. The semiquantitative screening of the antibacterial activity of the hydrogel 2.5 Se-HNF showed an inhibition of the microbial growth for all tested strains, the mean diameter of the inhibition zone being indicated in Table 6.

**Table 6 pharmaceuticals-17-00023-t006:** Semiquantitative screening of antimicrobial activity of 2.5 Se-HNF: average diameter of the inhibition zone (cm) ± standard error (SE).

Strain	*B. cereus*	*E. faecalis*	*S. aureus*	*C. albicans*
2.5 Se-HNF *	0.740 ± 0.003	0.760 ± 0.003	0.790 ± 0.010	0.810 ± 0.010
	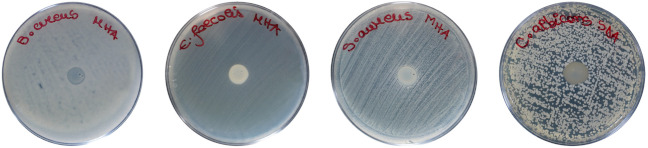			

* 2.5 Se-HNF—5% water-soluble chitosan in 0.4% never-dried bacterial nanocellulose enriched with 2.5 µg/mL SeNPsSb.

### 2.4. Physicochemical Characterization of Se-HNF

The contact angle of the hydrogel prepared with 2.5 µg/mL SeNPsSb embedded in the BNC-CS matrix (2.5 Se-HNF) was determined on two types of surfaces, on a polar surface (glass) and on a non-polar surface (polystyrene), and the results obtained were summarized in Table 7. The contact angle on the polar surface was smaller (52.82 ± 1.23°) than that obtained on the non-polar surface (73.85 ± 0.39°). The interfacial tension was 97.6 ± 0.47 mN/m.

Figure 8a shows the binding efficiency of 2.5 Se-HNF to mucin (%) determined by the periodic acid–Schiff (PAS) method. The mucin binding efficiency decreases as the amount of hydrogel and consequently the hydrogen bonds decrease (77.96 ± 1.97% at ratio 2.5 Se-HNF/mucin of 90 to 59.93 ± 3.7% at ratio 2.5 Se-HNF/mucin of 10). The SEM micrographs of 2.5 Se-HNF, mucin and their mixture (2.5 Se-HNF-Mu) shown in Figure 8b–d revealed a porous mesh-like structure of 2.5 Se-HNF (Figure 8b), which becomes a relatively compact structure upon mucin contact (Figure 8d).

The 2.5 Se-HNF hydrogel structure was investigated by FTIR spectroscopy in comparison with the IR spectra of chitosan and nanocellulose (Figure 9a). As can be seen, CS influence is dominant in 2.5 Se-HNF in all the regions of hydrogen bonds, C–H bonds, amide bands and polysaccharide bands, with particular signals marked around 3070 cm^−1^ for the N–H vibration in amide groups, -CH_3_ groups around 2922 cm^−1^ and C=O band around 1250 cm^−1^ in acetyl moieties. The BNC influence can be observed in the increased band of bound OH around 3333 cm^−1^ and the shift of the C–H band at 2886 cm^−1^ in Se-HNF, which is intermediate between CS (2870 cm^−1^) and BNC (2897 cm^−1^). The main interactions between NDBNC, CS and SeNPsSb within Se-HNF are the hydrogen bonds (H-bonds) between the proton-donor and functional groups containing non-bonding electrons, like oxygen in -OH, C=O and COOH and nitrogen in amines and amides. BNC and CS probably stabilize, additionally to the initial biocorona, the SeNPsSb via H-bonds and van der Waals interactions. These bonds and interactions induce band shifts in the FTIR spectrum of Se-HNF compared to the individual components, as visible in Figure 9a.

Figure 9b shows the interaction of the hydrogel with mucin by hydrogen bonds between amide groups and hydroxyl groups. There is a similitude between chitosan and mucin due to the acetyl groups, in particular through the amide band around 3070 cm^−1^, which is shifted upon interaction to a wavenumber higher than both mucin and hydrogel, methyl groups around 2924 cm^−1^ and C=O groups around 1240 cm^−1^. Actually, in 2.5 Se-HNF-Mu, two bands appear at 1244 and 1213 cm^−1^ for C=O groups bound to specific rings in mucin and chitosan. In mucin, the C=O is bound to a galactose ring and appears around 1232 cm^−1^ and is probably shifted to 1213 cm^−1^ in 2.5 Se-HNF-Mu. The C=O is bound to a glucose ring and appears around 1240 cm^−1^ in chitosan and around 1244 cm^−1^ in 2.5 Se-HNF-Mu. Mixing Se-HNF with mucin results in several band shifts at intermediary values between Se-HNF and mucin, suggesting multiple interactions at the molecular level dominated by hydrogen bonding.

The structure of the hydrogel was analyzed by XRD in comparison with the biopolymeric components, CS and BNC (Figure 9c). CS had the major influence in Se-HNF through the characteristic peaks at 2θ angles 12° and 23.04°. Depending on source and processing, CS usually shows two main diffraction peaks around 10° and 20° [35,36], but four peaks around 10.3°, 15.9°, 20.1° and 23° were also reported [37]. BNC mainly contributes with the diffraction peak from 22.66° characteristic for the stiff Iβ allomorphic structure consisting of two chains with monoclinic arrangement (PDF card No. 00-060-1502 cellulose Iβ, main peaks 14.83° for plane (−1,0,1), 16.42° for (1,0,1) and 22.71° for (0,0,2) diffraction plane). The Iα allomorph prevails in bacterial cellulose, some algae and vegetables and is more flexible due to the one-chain structure with triclinic arrangement, showing main diffraction peaks at 10.28° (0,0,1), 14.26° (1,0,0), 16.77° (0,1,0) and 21.80° (−1,1,0) (PDF card No. 00-056-1719). Additionally, amorphous cellulose consisting of disordered cellulose Iα and Iβ fragments shows main diffraction peaks at 15.28°, 19.78°, 27.13° and 36.10° (PDF card No. 00-060-1501) and might appear displaced in Se-HNF together with cellulose Iα due to interaction with CS, which gives small peaks at 8.66°, 17.24° and 19.12°. The 23.04° peak in hydrogel appears between CS (24.84°) and BNC (22.66°). The small peaks at 33.98° for BNC and 35.40° for CS disappear in the hydrogel due to rearrangement. The crystallinity of Se-HNF (59%) is intermediary between CS (46%) and BNC (84%). The diffractograms presented in Figure 9d mainly show that the interaction with mucin causes a decrease in crystallinity from 59% to 37% due to the lower crystallinity of mucin and a shift in all the diffraction peaks, suggesting a strong interaction of both CS and BNC with mucin.

The rheological analyses in oscillatory shear presented in Figure 10a reveal that the hydrogel exhibited viscoelastic behavior, with the loss modulus G″ higher than the storage modulus G′ up to the crossover modulus of 130.3 Pa at 21.22 rad/s, after which G′ became dominant. The storage modulus G′ represents the elastic component and describes the solid character of the formulation, while the loss modulus G″ represents the viscous component and describes the liquid character of 2.5 Se-HNF. The crossover modulus is defined as the intersection between the G′ and G″ curves, where G′ = G″ at the phase angle δ = 45°, tan(δ) = 1 and the angle correlates with the gel point after which the solid-like behavior becomes dominant (G′ > G″ at ω > 21.22 rad/s for Se-HNF). The complex viscosity η* of Se-HNF in Figure 10a decreased with the increase in ω up to a minimum of 8.59 Pa·s around ω = 25.12 rad/s, after which it increased in correlation with the increasing of the storage modulus after the crossover point. The phase angle δ expresses the viscoelastic behavior in terms of dominant elastic for values closer to δ = 0° (Hookean solid) and dominant viscous for values closer to δ = 90° (Newtonian liquid). In Figure 10a, the phase angle for Se-HNF decreases from 67.2° to 13.7°, suggesting the transition from liquid-like behavior to gel-like behavior with the increasing of ω on UP-ramp and the return to liquid-like behavior towards δ = 60.6° with the decreasing of ω on the reverse R-ramp. The stiffness k also starts to exponentially increase after the crossover point on UP-ramp and recovers with the decreasing of ω on R-ramp. The viscoelastic character defines, in our case, the friction between the polymer chains of nanocellulose and chitosan towards a gel-like structure by nanofibrils entanglement at ω > 21.22 rad/s, while the hysteresis loops of G′, G″, η*, δ and k variables evidenced the stability of Se-HNF from 0.1 to 100 rad/s and the reversibility of liquid–gel transition by nanofibrils’ disentanglement. At the end of the R-ramp, the slight increase in the η* and G′, together with the slight decrease in δ, suggest that the nanocellulose–chitosan fibrillar network rearranged in a slightly thicker structure after the oscillatory experiment.

The 3.5% mucin suspension showed in Figure 10b exhibited complex rheological behavior with two crossover points at 0.15 rad/s and 30.08 rad/s, respectively; between these two values, the elastic behavior is dominant, with G′ > G″. After the second crossover point at ω = 30.08 rad/s, the lack of G′ points together with the drastically increase in δ, k and η* suggest the breaking of the hydrogen-bonded protein macrostructure of mucin, which recovers on the R-ramp at lower ω.

The 1:1 dilution of Se-HNF with 3.5% aqueous mucin is presented in Figure 10 as the Se-HNF-Mu system, with the rheological experiments evidencing particular interaction aspects between the hydrogel and mucin. First of all, the oscillatory sweep mode depicted in Figure 10c evidenced that the initial crossover points of Se-HNF and Mu disappear in Se-HNF-Mu, suggesting that their interaction created a new macromolecular system without nanofibrils entanglement and with a dominant elastic behavior (G′ > G″) that increases with the increasing of ω, exponentially after ω > 3 rad/s.

The absence of entanglement of nanocellulose and chitosan fibrils in the presence of mucin suggests that protein fragments of mucin dispersed between BNC and CS nanofibrils and acted like a hydrogen-bonding nanofiller that induced an increased elasticity in the Se-HNF-Mu system. Secondly, the complex viscosity also shows a particular behavior for the Se-HNF-Mu system in comparison with individual Se-HNF and Mu fluids. The initial complex viscosity η* = 5 Pa·s of Se-HNF-Mu at small ω is lower than both Se-HNF η* = 32.4 Pa·s and Mu η* = 29.8 Pa·s and the minimum η* of Se-HNF-Mu is intermediary between those of Se-HNF and Mu (8.59 Pa·s for Se-HNF, 0.75 Pa·s for Se-HNF-Mu, 0.06 Pa·s for Mu), while the angular frequencies corresponding to the minimum η* varies in the order ω_min_ = 2.51 rad/s (Se-HNF-Mu) < ω_min_ = 25.12 rad/s (Se-HNF) < ω_min_ = 63.1 rad/s (Mu). The fact that the Se-HNF-Mu system has a minimum η* at lower ω than the crossover frequency ω = 25.12 rad/s of Se-HNF suggests the involvement of new intermolecular bonds between Mu, BNC and CS, stronger than the hydrogen bonds in mucin alone but weaker than the entanglement forces of nanocellulose–chitosan nanofibrils, which could be assigned to a combination of hydrogen bonds and mucin fragments intercalation between the BNC-CS nanofibrils.

The shear behavior of the hydrogel presented in Figure 10d is particular for a shear thinning behavior, since the viscosity decreased from 26 Pa·s to 2.55 Pa·s with the increasing of the shear rate from 1 to 300 s^−1^. The low value of thixotropy and the similitude between the UP-ramp and R-ramp suggests that the hydrogel is stable even up to 300 s^−1^, which corresponds to high brushing forces [38]. The stress curve of Se-HNF was best fitted with the power-law model of Ostwald de Waele, confirming the shear thinning behavior. In Figure 10e, it can be observed that mixing the hydrogel with the mucin solution leads to a lower initial viscosity of only 1 Pa·s and an increased thixotropy compared with the hydrogel, together with the transformation into a Herschel–Bulkley fluid with yield stress, suggesting that it will not easily flow under gravity unless the yield stress is exceeded. The yield stress is induced by mucin that has a low yield stress, fitted and determined with the Casson model, and also has a higher rheopectic (anti-thixotropic) behavior with the stress curve on the R-ramp than on the UP-ramp.

The rheological experiment in axial mode is the most relevant experiment for the mucoadhesion properties because it measures the axial force opposing the detachment of the test cylinder and can thus be considered a determination of the mucoadhesion force. The axial force is higher in 2.5 Se-HNF (0.186 N) compared to 2.5 Se-HNF-Mu (0.079 N), suggesting a higher surface energy of adhesion for 2.5 Se-HNF (Figure 10g–i). The mucin alone has an adhesion force F = 0.272 N higher than the Se-HNF hydrogel, but an adhesion time of only 1 s. The adhesion time of 2.5 Se-HNF and 2.5 Se-HNF-Mu was 34.03 s and 5.05 s, respectively. If we consider the adhesion energy as the work of adhesion necessary to separate 1 m^2^ of joined materials for the work distance of separation, we can estimate the adhesion energies of Se-HNF, Mu and Se-HNF-Mu systems by interpolating the distances in the axial force graphic at the value of null axial force; the corresponding determined distances (d) for Se-HNF, Mu and Se-HNF-Mu are 2.081, 7.946 and 1.360 mm. The contact surface is the 40 mm geometry (cylinder) area, meaning S = 1.257 × 10^−3^ m^2^. The resulting adhesion energies calculated with the formula A_E_ = F × d/S were 0.308, 1.719 and 0.085 J/m^2^ for Se-HNF, Mu and Se-HNF-Mu, respectively. If the energy of adhesion represents the adhesion with the external surfaces, then the difference between the adhesion energy of Se-HNF and the Se-HNF-Mu can be considered the cohesion energy, specifically, the internal energy of Se-HNF bonds with mucin. In our case, the internal energy would be 0.223 J/m^2^, which is 72.4% from the initial adhesion energy of Se-HNF and suggests a strong interaction between Se-HNF and Mu. Additionally, the axial detachment curves in Figure 10g–i were fitted with an exponential function, the most relevant parameter being the power exponent c, which can be further interpreted as the speed of detachment. Consequently, the highest adhesion property can be correlated with the lowest speed of detachment, respectively, e.g., c = 21.02 for mucin. The lower detachment speed of Se-HNF c = 61.18 compared with c = 74.88 for Se-HNF-Mu confirms that Se-HNF has a higher adhesive character before the contact with mucin and that it converts a main part of its adhesive energy through BNC and CS into cohesive energy with mucin.

## 3. Discussion

We reported in this study the synthesis and biological and physicochemical properties of a mucoadhesive selenium-enriched hydrogel nanoformulation (Se-HNF) intended for the health of the oral cavity. The Se-HNF hydrogel is structurally based on biocompatible never-dried bacterial nanocellulose (NDBNC) from kombucha fermentation for its hydrophilic flexible network of nanofibrils and on soluble fungal chitosan (CS) for its mucoadhesive and antimicrobial properties. Additional antioxidant, antimicrobial and antibiofilm activities are provided by biogenic selenium nanoparticles (SeNPs) with a particular biocorona resulted from the phytosynthesis process in an aqueous extract of sea buckthorn leaves (SeNPsSb), as further discussed. As we already mentioned, to the best of our knowledge, this is the first study on SeNPs phytosynthesized by an aqueous extract of sea buckthorn leaves and on this type of formulation.

Sea buckthorn leaves have gained more interest in recent years as they turned out to be a valuable source of bioactive compounds. However, the amount of bioactive compounds may vary depending on the cultivar, origin, harvesting time and the extraction process. The TPC of SbLEx in this study, i.e., 46.59 ± 0.93 mg GAE/g dry weight (Table 1), is in agreement with another study, which reported a phenolic content of sea buckthorn aqueous extract of 40.49 ± 2.10 mg/g dry leaf [7]. The previously reported TFC varied between 0.83 and 2% depending on the subspecies and origin, and, in some varieties, the flavonoid content was undetectable [39]. The phenolic acid content has been reported from 4511.4 to 6150.7 µg/g, depending on cultivars [40]. The compounds identified by LC-MS analysis in the present study (Table 2) have been previously reported in sea buckthorn leaf extracts. GABA, which is an ubiquitous non-proteinogenic amino acid in plants, was identified in sea buckthorn leaf extract by ^1^H and 2D nuclear magnetic resonance (NMR) [41]. Quinic acid is a compound that was identified in various plants, including in sea buckthorn leaves [42,43]. Together with caffeic acid, it is a structural component of chlorogenic acid [44]. Cirsium aldehyde results from a biosynthesis process, i.e., by etherification of two molecules of 5-hydroxymethylfurfural, and it is a natural product which was also found previously in sea buckthorn [45]. Thiamine (vitamin B1) has been identified in sea buckthorn leaves with a content of 1.4 ± 0.1 mg × kg^−1^. Quercitrin is a flavonoid found in many plants which has been reported between 0.17 ± 0.01 and 1.84 ± 0.05 mg/g in sea buckthorn leaf tea depending on manufacturer and factory time [46]. It has antioxidant and anti-inflammatory activity [47]. Quercetin-7-rhamnoside results from the addition of the rhamnosyl group to the 7-OH group of quercetin [48]. It was previously identified in sea buckthorn leaf tea by LC-MS analysis, along with astragalin [42], a flavonoid reported as present in various parts of several plants. Astragalin has often been reported in pharmacological studies for its remarkable antioxidant, anti-inflammatory, anti-tumor and neuroprotective properties, but it has proven its effectiveness even in the cosmetic field due to its collagenase-inhibiting action, as well as its protective effect against UV radiation [49]. Isorhamnetin-3-O-rutinoside, along with isorhamnetin-3-O-glucoside, have been reported in sea buckthorn cultivar leaf powders with a content that varies between 479.3 ± 16.31 and 746.4 ± 25.56 µg/g dry weight, respectively, between 256.8 ± 8.61 and 410.9 ± 13.94 µg/g dry weight [40]. In another study, isorhamnetin-3-O-rutinoside varied between 38.7 and 84.0 mg/100 g dry weight, while isorhamnetin-3-O-glucoside varied between 7.61 and 26.0 mg/100 g dry weight, depending on subspecies. Together with the aforementioned compounds, isorhamnetin-3-O-glucoside-7-O-rhamnoside varied between 112 and 187 mg/100 g dry weight [50]. Prodelphinidins, also known as proanthocyanidins, are oligomers or polymers of epicatechin and epigallocatechin subunits. Prodelphinidin trimer with the assigned structure epicatechin-epigallocatechin-epigallocatechin has been reported in sea buckthorn [51]. Quercetin 3-O-(6-O-feruloylglucoside)-glucoside-7-O-rhamnoside is a flavonol glycoside which was also identified in sea buckthorn leaves [52]. The presence of catechin (C) in sea buckthorn leaves has been reported with values between 115.7 and 2030.0 mg × kg^−1^ [53] depending on cultivars and ripening times and for epicatechin (EC) between 108.5 and 247.6 mg × kg^−1^ depending on harvesting time [54]. The rutin (Que-rut) content from *Hippophae rhamnoides* cultivar leaf powders varied within the range 310.2–576.1 μg × g^−1^ [40] and even up to 0.3% [55].

Several attempts to use an eco-friendly reducing agent (ascorbic acid, sugars) for the production of SeNPs with different precursors (selenium oxo-acid salts) have been reported [30,31,56,57,58,59,60], but most of the time, these SeNPs are not stable in aqueous solutions, which is an important factor in their bioavailability and bioactivity. Therefore, due to the rich phenolic content of various plant extracts along with their ability to be both reducing and stabilizing agents in the green synthesis of SeNPs, plant extracts have been increasingly used to obtain SeNPs with unique properties and diverse biological activity [61]. The size of the nanoparticles, as well as their homogeneity, is an important aspect in the biomedical field. Our results highlighted polydisperse SeNPs with a PDI of 0.34 a.u. and size between 10 and 150 nm, probably including aggregates of small NPs. The size distribution from TEM was similar to that resulting from SAXS analysis, with a main diameter of 15–25 nm, reflecting the diameter of the selenium core. The DLS analysis indicated a higher main diameter of 100 nm, reflecting the contribution of the biocorona and hydration layer. By using the aqueous extract of *Clausena dentata*, SeNPs with sizes between 46.32 and 78.88 nm were previously obtained [62]. In another study, SeNPs with sizes between 50 and 150 nm were produced with 1% *Trigonellafoenum graecum* (fenugreek) extract, 40 mM ascorbic acid and 30 mM selenious acid [63]. By using an aqueous extract from *Terminalia arjuna* leaves, polydisperse SeNPs with sizes between 10 and 100 nm were biosynthesized [64]. Many other plant extracts have been used in the production of unique SeNPs like *Citrus lemon* leaf extract [32], *Psidium guajava* (guava) alcoholic leaf extract [65], *Vitis vinifera* (raisin) fruits extract [66] and *Capsicum annuum* L. extract [67]. SeNPs were also phytosynthesized using *Pseuderanthemum palatiferum* leaf extract and, via DLS analysis, it was found that the average diameter of the SeNPs was 247 nm, while the Zeta potential analysis indicated a high stability (−29 mV) [68]. In the present study, SeNPsSb presented the same high degree of stability and negatively charged surface, i.e., −29.98 ± 0.421 mV (Appendix A).

Bacterial plaque is the main factor involved in the etiology of periodontal disease. It has been defined as a diverse community of microorganisms which are localized on the teeth surface, embedded in a polymeric matrix (e.g., polysaccharides, proteins and glycoproteins) derived from saliva deposits or produced by bacteria [2,69]. The main nutrients for plaque-forming bacteria are produced from the catabolism of endogenous nutrients. Since pathogens can modulate the host response, shifting the balance from homeostasis to dysbiosis and leading to dysbiotic film development, new antimicrobial strategies are emerging in periodontal disease therapy, which are also essential for countering the global phenomenon of multidrug resistance (MDR), including bacteria/antibiotic resistance (AR) [70].

*B. cereus* is a Gram-positive bacteria which is widely distributed in the environment. It is one of the most common bacteria responsible for food contamination due to its ability to withstand extreme temperatures. Although there is not yet much knowledge on its involvement in periodontal diseases, it was reported that 19 strains of *B. cereus* and *B. thuringiensis* were isolated from the oral cavity in marginal and apical periodontitis [71]. Our results indicated the efficiency of SeNPsSb in inhibiting *B. cereus* growth. The semiquantitative screening of antimicrobial activity indicated an inhibition of *B. cereus* growth at 0.5 mg/mL SeNPsSb (Table 5). Also, the MIC value was 1.5 mg/mL, while the IC50 value was 0.708 ± 0.031 mg/mL (Table 6).

*E. faecalis* is a Gram-positive bacteria which was also isolated from the subgingival biofilm of gingivitis (5.9%) and periodontitis (41.7%) diagnosed patients. Enterococci surface protein (ESP) was also detected in various oral infections, as well as secretory metalloprotease gelatinase E (gelE), associated with *E. faecalis* pathogenesis [72]. SeNPsSb were effective against *E. faecalis* growth. The semiquantitative analysis indicated a dose-dependent inhibition of microbial growth, with the diameter of the inhibition zone increasing from 0.39 ± 0.001 cm at 0.1 mg/mL SeNPsSb to 0.51 ± 0.006 cm at 0.2 mg/mL and reaching 0.86 ± 0.009 cm at 0.5 mg/mL without significant increases at higher doses of SeNPsSb (Table 5). The MIC value was 1 mg/mL, while the IC50 value was recorded at a relatively low concentration, i.e., 0.039 ± 0.0082 mg/mL (Table 6). Another important aspect was the significant effect of SeNPsSb against the formation of *E faecalis* biofilm.

*S. aureus* is a Gram-positive bacteria included in the ESKAPE group (*Enterococcus faecium*, *Staphylococcus aureus*, *Klebsiella pneumoniae*, *Acinetobacter baumannii*, *Pseudomonas aeruginosa*, *Enterobacter* sp.) of multidrug-resistant bacteria, responsible in many cases for untreatable infections with the available antimicrobial agents. Staphylococcus species were found in approximately 50% of gingivitis and periodontitis cases [73]. Using the semiquantitative assay, inhibition of *S. aureus* was found at a concentration of 0.05 mg/mL, and no change in halo diameter was observed when increasing the concentration of nanoparticles. On the other hand, the quantitative analysis revealed an interesting effect of our SeNPsSb. The MIC value of SeNPsSb was 1 mg/mL, while the IC50 value was observed both at relatively high concentrations of SeNPsSb, as well as at low concentrations. By analyzing the range 2–0.1 mg/mL SeNPsSb, the IC50 was 0.288 ± 0.004 mg/mL (Table 6); then, from 0.05 to 0.0005 mg/mL, the *S. aureus* inhibition starts to increase up to 50.23 ± 1.5% at 0.01 mg/mL and 56.17 ± 1.8 mg/mL at 0.005 mg/mL (Figure 5c). Therefore, in this case we have a dual effect of our SeNPsSb. Lately, the hormetic effects of selenium (Se) have been reported as having certain properties at both high and low doses [74]. The main organic forms of Se are seleno-amino acids (selenomethionine—SeMet and selenocysteine—SeCys), which are incorporated into selenopeptides and selenoproteins [10], selenium being an analogue of sulfur. It has been reported that until a nutritional plateau is reached, Se is specifically incorporated into selenoproteins, while at higher concentrations, excess Se becomes toxic through nonspecific incorporation, which leads to the formation of selenylated proteins that destabilize reactive oxygen species (ROS) homeostasis [74]. By colony-forming unit (CFU) analysis after SeNPs treatment, it was reported in a previous study that in the case of *S. aureus*, the CFU at 0 µg/mL SeNPs was 68.0 ± 11.5 and decreased to 57.8 ± 13.4 after 0.5 µg/mL SeNPs treatment, followed by an increase up to 74.4 ± 9.8 at 1 µg/mL SeNPs, with a subsequent decrease down to 51.8 ± 11.3 at 2.5 µg/mL SeNPs, then reaching 63.0 ± 17.3 at 5 µg/mL SeNPs with a significant decrease down to 39.2 ± 5.7 at 10 µg/mL [75]. However, even if there were some interesting variations, the effect noted in the aforementioned research was not statistically significant up to 10 µg/mL SeNPs.

The most common oral infections have been reported to be caused by *C. albicans*, a pathogenic fungus [76]. Moreover, numerous studies have indicated the presence of *C. albicans* in subjects with periodontitis [77]. Despite the fact that our phytosynthesized SeNPsSb had no effect against *C. albicans* even at the highest tested dose, i.e., 2 mg/mL, due to the well-known antimicrobial potential of chitosan [25], the Se-HNF formulation was effective against all strains, including *C. albicans*. HNF suspensions showed high biocompatibility on gingival fibroblasts HGF-1 even at 1 mg/mL concentration, but there was no potential to inhibit the production of ROS. By adding 0.5 µg/mL SeNPsSb in the HNF suspensions (0.5 Se-HNF), a high degree of cytocompatibility and in vitro antioxidant activity were observed, but the ROS level was still above the level recorded in the case of the negative control (untreated cells). Only by adding 2.5 µg/mL SeNPsSb in the HNF suspensions (2.5 Se-HNF) were both a high degree of cytocompatibility and the potential to inhibit the ROS production at the level of the negative control, or even below, obtained. By varying the microbial density in order to determine the potential of Se-HNF to prevent microbial infections, it was observed that increasing the concentration of SeNPsSb from 0.5 µg/mL to 2.5 µg/mL, there was a significant decrease in the microbial growth compared to both the HNF hydrogel suspension and 0.5 Se-HNF. Together with the antibiofilm effect of SeNPsSb on *E. faecalis*, we can conclude that SeNPsSb and HNF complement each other.

The salivary mucin content measured in healthy subjects is about 1.2 mg/mL, in addition to other proteins and mineral salts, with the main salivary glycoproteins being MUC7 and MUC5B. The viscoelastic character of saliva or other mucilaginous substances is essential for the well-being of the host because they act as a barrier, delaying the diffusion of different compounds secreted by the pathogens and even encapsulating foreign agents [78]. Moreover, it has been reported that the mucin content tends to increase in periodontal disease [79], this effect being determined by the action of sialidases secreted by *Tannerella forsythia*, which cleaves the sialic acid from the mucin structure, leading to the degradation of the physiological and immunological barrier located on the surface of the oral epithelium. Our results indicated a significant mucin binding efficiency of 77.96 ± 1.97% at the ratio 2.5 Se-HNF/mucin of 90, which decreased as the amount of hydrogel and consequently the hydrogen bonds decreased. The important role of hydrogen bonds made by amino donor groups, as well as the role of electrostatic interactions of positivized amino groups with negatively charged macromolecules in the mucin structure, mainly sialic acid and ester sulphates, was highlighted by the reduction in amino groups [80]. The contact angle measurement highlighted an amphiphilic behavior with a more hydrophilic rather than hydrophobic character of the Se-HNF nanoformulation, with the semi-acetylated chitosan having an increased solubility in water (up to pH 7.4) compared to the nonmodified chitosan [80].

The nanocellulose–chitosan biopolymeric matrix in which the SeNPs were embedded behaves like a soft uncross-linked hydrogel with 95.18% water content that further dilutes in contact with saliva and gingival mucin, simulated as a 50% dilution with a 3.5% mucin aqueous solution, without losing its structural stability, as evidenced by rheology. The Se-HNF hydrogel has a viscous flowing behavior (loss modulus G″ > storage modulus G′) at angular frequencies lower than the crossover frequency of 21 rad/s; over this frequency, the elastic behavior becomes dominant with G′ > G″ as a strong structural hydrogel. The rheological behavior in the strain oscillation mode was performed between 0.1 and 100 rad/s: the range frequencies between 0.1 and 1 rad/s are related to draining under gravity and storage stability, the frequencies between 1 and 10 rad/s are related to transportation stability and laminar flowing like in the inversion of a tube or recipient, while the frequencies between 10 and 100 rad/s describe the usual range for manual application of cosmetics, from gentle to energetic spreading of formulations [38]. In our case, the ω values between 10 and 50 rad/s can be practically related to hydrogel spreading from the tube to the tooth brush or other application tool, followed by gentle spreading of the hydrogel on sensitive gums, while higher frequencies from 50 to 100 rad/s can be related to energetic brushing to remove bacterial plaque (dysbiotic biofilm). Considering these aspects, the Se-HNF hydrogel will behave like a soft gel during gentle application, while during energetic brushing with frequencies higher than the crossover frequency of 25 rad/s, the hydrogel elastic behavior and stiffness increase, which will facilitate the removal of dysbiotic biofilm. The G′ and G″ values for Se-HNF ranged between 1 and 1000 Pa in the full ω range 0.1–100 rad/s, somewhat comparable with two injectable hydrogels based on glycol–chitosan, poly(ethylene oxide) and alginate for dental pulp regeneration [81] but more viscous than the dominant elastic injectable hydrogels previously mentioned. Also, in comparison with the previously mentioned study, the complex viscosity ranges between 32 to 8 Pa·s for the Se-HNF hydrogel and between 1 and 10 Pa·s after the contact with diluted mucin, while for the ionic crosslinked glycol–chitosan scaffolds, η* varied between 10^5^ and 10 Pa·s [81].

The adhesion force and energy determined for the Se-HNF hydrogel, F = 0.186 N and A_E_ = 0.308 J/m^2^, respectively, were lower than the values of two binary hydrogels based on BNC and poloxamer (F = 0.34–0.38 N, A_E_ = 1.2 J/m^2^) and a ternary formulation with BNC, poloxamer and high-molecular-weight chitosan (F = 1.7 N, A_E_ = 9.1 J/m^2^) [82]. The values can also be considered 10 times lower than the ones obtained for a strongly adhesive alginate–catechol-peptide hydrogel designed to treat the periimplantitis disease, which had values in the range of F = 1.8 N and A_E_ = 4 J/m^2^ [83]. The Se-HNF hydrogel is not intended for a long residence time; thus, for the intended anti-plaque application, it can be considered suitable.

## 4. Materials and Methods

### 4.1. Materials

Sea buckthorn leaves (*Elaeagnus rhamnoides*, syn. *Hippophae rhamnoides*, cv. Mărăcineni) were collected from Cooperative Biocătina, Găești, Romania. The Symbiotic Culture of Bacteria and Yeast (SCOBY) originated from a Romanian culture [18]. Gingival fibroblasts (HGF-1, ATCC CRL-2014) were used for biocompatibility assays. Antimicrobial activities were carried out using the following strains: *Bacillus cereus* NCTC 10320, *Enterococcus faecalis* ATCC 29212, *Staphylococcus aureus* ATCC 25923 and *Candida albicans* ATCC 10231. The following chemicals were used: Chitoly^®^ OM oyster mushroom chitosan (Handary, Brussels, Belgium), ascorbic acid, sodium selenite, mucin from porcine stomach type II, 1,1,3,3-tetra-ethoxypropane 99%, 2-thiobarbituric acid ≥ 98%, gallic acid, quercetin dihydrate, neocuproine, 2,2-diphenyl-1-picrylhydrazyl, Dulbecco’s Modified Eagle’s Medium low glucose, D-(+)-glucose, sodium bicarbonate, trypsin from porcine pancreas, dimethyl sulfoxide 99.5%, 4′,6-diamidino-2-phenyindole, antibiotic antimycotic solution 100× stabilized, dilactate sodium deoxycholate (Sigma Aldrich, St. Louis, MO, USA), Cell counting kit-8 (Bimake, Houston, TX, USA), absolute ethanol 99.5%, hydrochloric acid 37%, acetic acid (Chimopar Srl, Bucharest, Romania), Viability/Cytotoxicity Assay Kit (Biotium, Fremont CA, USA), Trolox 97% (Acros Organics, Thermo Fisher Scientific, Pittburghs, PA, USA), Folin–Ciocalteu’s phenol reagent, iron chloride (III) (Merck, Darmstadt, Germany), di-sodium hydrogen phosphate dihydrate, sodium dihydrogen phosphate monohydrate, tris-(hydroxymethyl)-aminomethane, potassium iodide, sodium chloride, paraformaldehyde, Triton X-100, trichloroacetic acid for analysis, hydrogen peroxide 30%, hydrochloric acid 1N, sodium acetate, sodium hydroxide, sodium sulfide, sodium nitrite, sodium molybdate, Müeller–Hinton agar, Sabouraud dextrose agar, Müeller–Hinton broth, Sabouraud dextrose broth, potassium iodide, fuchsin basic for microscopy, crystal violet (Scharlau, Barcelona, Spain), 2,4,6-tri (2-pyridyl-1,3,5-triazine) 98%, sodium n-dodecyl sulfate 99%, aluminum chloride, hydroxylamine hydrochloride (Alfa Aesar, Haverhill, MA, USA), Phalloidin-iFluor 488Reagent (Abcam, Cambridge, United Kingdom), albumin bovine fraction V, pH 7.0 (Janssen Chimica, Beerse, Belgium), FBS USDA APPD. ORIGIN (Thermo Fisher Scientific, Waltham, MA, USA), chlorogenic acid, 2,7-dichlorodihydrofluorescein (Cayman Chemicals, Ann Arbor, MI, USA), glacial acetic acid, sodium chloride, activated charcoal, potassium metabisulfite (Chimreactiv, Bucharest, Romania), periodic acid (VWR International, Radnor Twp, PA, USA) and methanol (Honeywell, Wabash, IN, USA).

### 4.2. Phytosynthesis of SeNPs with Sea Buckthorn Leaves Aqueous Extract (SbLEx)

A ratio of 1:10 (*w*/*v*) sea buckthorn leaves to double-distilled water was used in order to prepare the sea buckthorn leaf aqueous extract (SbLEx). The mixture was kept in an ultrasonic bath for 30 min at 30 °C and centrifuged for 30 min at 7500 RCF (Universal 320R Centrifuge Hettich, Tuttlingen, Germany). Subsequently, a solution of 100 mM ascorbic acid followed by the SbLEx solution were added dropwise into a 10 mM Na_2_SeO_3_ solution under stirring, in a final ratio of 0.5:1:1 (*v*/*v*/*v*). The Na_2_SeO_3_ and ascorbic acid solutions were prepared in double-distilled water (ddH_2_O). After 1 h, the mixture was ultracentrifuged at 210,000 RCF at 10 °C (CP100NX Ultracentrifuge, Hitachi Koki, Tokyo, Japan), and the sediment was washed three times with Milli-Q water using the previously mentioned ultracentrifugation parameters. Afterwards, the SeNPsSb were freeze-dried (ScanVac CoolSafe 55-4 freeze-dryer, LaboGene, Bjarkesvej, Denmark) at −55 °C freeze-drying temperature.

#### 4.2.1. Determination of Se Conversion Yield (%)

The total Se^0^ content was assessed by subtracting the Se^0^ content of the supernatant obtained by ultracentrifugation of the final suspension at 210,000 RCF from the Se^0^ content of the final suspension. For the determination of the SeNPsSb yield, the method of Biswas et al. [84] with some modifications was employed. The content of Se^0^ was determined from the sediment containing SeNPsSb resulting after the ultracentrifugation and washing described above in Section 4.2, after resuspending it in the same volume as before ultracentrifugation. A stock solution of 0.1 M Na_2_SeO_3_ was used for the calibration curve prepared in the concentration range of 2–20 mM. The standards for the calibration curve, the initial selenate solution and the sample were processed in the same way: 1 mL of the sample/standard was mixed with 0.1 mL of 50 mM NH_2_OH to reduce selenite to Se^0^; the samples were subjected to N_2_ drying with a concentrator MultiVap 54 (LabTech, Sorisole, Italy) and the pellet was resuspended in 0.2 mL 1 M Na_2_S. Although Se was already reduced in the sediment sample, it was treated with NH2OH as well because Se in the nanoparticles have the tendency to undergo disproportionation in solution, especially in the absence of a reducing agent. The resulting solutions were incubated for 1 h at room temperature and the absorbance was measured in triplicate at λ = 500 nm using a microplate reader (CLARIOstar BMG Labtech, Ortenberg, Germany). The Se conversion yield was determined by Equation (1):(1)$$\eta =\frac{C_{s}}{C_{i}}\times 100$$
where *η* is the yield, *C_s_* is the Se^0^ content in the sediment sample after ultracentrifugation and *C_i_* is the initial selenite content determined by the same method. The yield was determined in triplicate and the standard deviation was calculated.

### 4.3. Hydrogel Preparation

The BNC-CS matrix without SeNPsSb (HNF) was prepared by adding 5% (*w*/*v*) water-soluble chitosan in a suspension of 0.4% (*w*/*v*) never-dried bacterial nanocellulose (NDBNC) in ddH_2_O, prepared according to a protocol previously published [27]. An Ultra-Turrax homogenizer (Ultra-Turrax^®^, IKA, Staufen, Germany) was used to homogenize the mixture.

The biopolymeric hydrogel enriched with SeNPsSb (Se-HNF) was prepared by adding 2.5 µg/mL of SeNPsSb in a suspension of 0.4% (*w*/*v*) NDBNC. Subsequently, 5% water-soluble chitosan was added and the homogenization was carried out as in the case of HNF, resulting in the sample 2.5 Se-HNF.

In order to study the interaction of Se-HNF with mucin, we prepared a 3.5% mucin suspension in ddH_2_O, which was left to homogenize at room temperature in a Loopster digital rotating shaker (IKA, Staufen, Germany), resulting in the sample Mu. Afterwards, 2.5 Se-HNF was mixed with the mucin suspension using a ratio of 1:1 (*v*/*v*), resulting in the sample Se-HNF-Mu.

These samples were analyzed in their initial, original state, except for the analyses where it is mentioned that they were freeze-dried. For the freeze-drying, we used a ScanVac CoolSafe 55-4 freeze-dryer (LaboGene, Bjarkesvej, Denmark) at −55 °C freeze-drying temperature.

### 4.4. Physicochemical Characterisation

#### 4.4.1. Determination of Total Polyphenol Content (TPC) of SbLEx

To determine TPC, 10 μL of the sample was pipetted into a 96-well plate, over which 90 μL of bidistilled water and 10 μL of Folin–Ciocalteu reagent were added. After 5 min of shaking, we added 100 μL of 7% Na_2_CO_3_ and 40 μL of bidistilled water. The plate was incubated for 60 min at room temperature and the absorbance of the samples was read at λ = 765 nm in triplicate. Using the same steps as for the samples, the calibration curve was performed in the concentration range of 0–250 µg × mL^−1^ starting from a stock solution of 500 µg/mL gallic acid in 70% EtOH [27,85,86].

#### 4.4.2. Total Flavonoid Content (TFC) Measurement of SbLEx

In order to assess TFC, 25 µL of 10% sodium acetate was mixed with 25 µL of the sample, 30 µL of 2.5% AlCl_3_ and 170 µL of double-distilled water. The absorbance was measured at λ = 430 nm in triplicate using a microplate reader after 45 min of incubation in the dark at room temperature. A stock solution of 500 μg/mL quercetin in 70% EtOH was used for the calibration curve, which was run in the concentration range of 0–100 µg × mL^−1^, using the same steps as for the samples [27,86,87].

#### 4.4.3. Assessment of Total Hydroxycinnamic Acid Content (HAT) of SbLEx

For HAT determination, 25 µL of the sample was mixed with 50 µL of 0.5 M HCl, 50 µL of 1%, 1% solution of sodium nitrite–sodium molybdate (*w*/*v*), 50 µL of 8.5% NaOH (*w*/*v*) and 75 µL of double-distilled water. After shaking, the absorbance of the samples was read in triplicate at λ = 524 nm using a microplate reader. A stock solution of 1 mg/mL chlorogenic acid in 70% EtOH was used for the preparation of the calibration curve in the concentration range of 0–300 µg × mL^−1^ [27,86].

#### 4.4.4. Quantification of Total Anthocyanins Content (TAC) of SbLEx

TAC was assessed by mixing 1 mL of SbLEx with 1.5 mL of 0.025M potassium chloride solution (pH = 1) and by mixing the same volume of the sample with 1.5 mL of 0.4 M sodium acetate buffer (pH = 4.5). The absorbance (A) was measured after 30 min of incubation at room temperature with a spectrophotometer (Ocean Optics UV-VIS-NIR DH-2000-BAL, Orlando, FL, USA) at two different wavelengths, namely 520 and 700 nm. The TAC was calculated according to [27].

#### 4.4.5. Liquid Chromatography–Time-of-Flight/Mass Spectrometry Analysis (LC-TOF/MS) of SbLEx

LC-TOF/MS (Agilent Technologies 1200 Infinity Series LC system, Santa Clara, CA, USA), controlled by the B.06.01 Mass Hunter Acquisition software, was used for SBLEx analysis. The chromatographic separation was carried out using an Agilent Poroshell 120 SB-C18 column (2.7 μm particle size, 4.6 inner diameter × 50 mm length). The mobile phase was prepared by mixing 80% acetonitrile and 20% water containing 0.1% trifluoroacetic acid. A flow rate of 0.3 mL × min^−1^ was used. The ESI was operated in the positive mode with the following parameters: dual spray needles, 3.0 L × min^−1^ drying gas, 15 psig nebulizer pressure, 3500 V capillary voltage and 175 V fragmentor voltage. The mass range of data acquisition was 100–1000 *m*/*z*.

#### 4.4.6. High-Performance Liquid Chromatography (HPLC-DAD) Analysis of SbLEx

The chromatographic analysis of catechin (C), epicatechin (EC) and quercetin 3-rutinoside (Que-rut) from SbLEx was performed on a HPLC-DAD Agilent 1100 system with the method described by [88] on a Kromasil 100-5C18 column (150 mm × 4.6 mm, 5 μm). A stock solution of 100 mg × L^−1^ was prepared in methanol and the calibration standards were obtained by appropriate dilution from stock solution. The peak areas of analytes versus concentrations were found to be linear (Appendix A).

#### 4.4.7. Evaluation of Antioxidant Activity

DPPH assay: a stock solution of 0.3 mM DPPH in absolute ethanol was prepared for the determination of the antioxidant activity by the DPPH method. The SbLEx sample was mixed 1:1 with the DPPH stock solution (100 µL SbLEx with 100 µL DPPH stock solution) in triplicate. The absorbance was measured at λ = 517 nm using a microplate reader after 30 min of incubation at room temperature in the dark. A stock solution of 1 mM Trolox prepared in 70% EtOH was used for the calibration curve performed in the concentration range 0–0.15 mM [27,86].

FRAP assay: the FRAP reagent was prepared by mixing 300 mM sodium acetate buffer, pH = 3.6, 10 mM TPTZ and 20 mm FeCl_3_ (10:1:1—*v*/*v*/*v*). It was incubated at 37 °C (MIR-154-PE, PHCBi Panasonic, Osaka, Japan) prior to sample analysis. To 15 µL of the sample/standard, 285 µL FRAP reagent was added. The samples were incubated at 37 °C in the same incubator in the dark for 30 min. The calibration curve was performed in the concentration range of 0–450 µM Trolox, starting from a stock solution of 10 mM Trolox in 70% ethanol [27,86].

CUPRAC assay: the reaction mix included 10 µL of the sample, 30 µL of 5 mM CuSO_4_, 30 µL of 3.75 mM neocuproine and 280 µL of double-distilled water. The absorbance of the samples carried out in triplicate was measured after 30 min of incubation at λ = 450 nm with a microplate reader. A stock solution of 10 mM Trolox in 70% ethanol was used for the calibration curve prepared in the concentration range of 0–2 mM [27,86,89].

In the case of SeNPsSb, we worked in Eppendorf tubes, and before the absorbance reading, the samples were centrifuged at 6000 RCF and the supernatant was transferred in 96-well plates. The SeNPsSb at the tested concentrations (100, 200, 500 µg/mL) in double-distilled water without the reaction mix were used as blanks, with the resulting optical densities being subtracted from the optical densities of the samples.

#### 4.4.8. Fourier Transform Infrared Spectroscopy (FTIR) Analysis

The FTIR spectra of freeze-dried samples were recorded in Attenuated Total Reflectance (ATR) mode using an IRTracer-100 spectrometer (Shimadzu, Kyoto, Japan). The analyses were assessed as mean of 45 scans with a resolution of 4 cm^−1^ in the mid-IR spectral range of 4000–400 cm^−1^. OriginPro software version 9.9.5 from OriginLab Corporation was used in order to create the graphs (Northampton, MA, USA).

#### 4.4.9. X-ray Diffraction (XRD) Analysis

XRD analysis of freeze-dried samples was performed using a Rigaku-SmartLab diffractometer (Rigaku, Tokyo, Japan), working with 45 kV voltage, 200 mA intensity and Cu_Kα1_ incident radiation at 1.54059 Å wavelength in wide-angle X-ray scattering (WAXS) for 2θ values between 2 and 90°, with 0.02° resolution and 4°/min scanning speed. The SeNPsSb nanoparticles were additionally analyzed by small-angle X-ray scattering (SAXS) in the 2θ range 0–1.5°, corresponding to scattering wavevector q range 0–1.068 nm^−1^. The particle diameter was estimated with the formula d = 2π/q, and the scattering vector q with the formula q = 4π ∙ sin(θ)/λ, where λ = 1.54059 Å [90,91]. The identification of characteristic peaks and determination of the crystallinity of the samples were performed with the PDXL 2.7.2.0 software and the final figures were obtained using the OriginPro 9.9.5. software.

#### 4.4.10. Transmission Electron Microscopy (TEM)–Energy Dispersive X-ray (EDX) Analysis

SeNPsSb were visualized by TEM (TECNAI F20 G2 TWIN Cryo-TEM (FEI) transmission electron microscope, Houston, TX, USA). After adding 10 μL of the sample on a carbon type-B, 200 mesh copper grid (Ted Pella, Redding, CA, USA), the excess liquid was removed with filter paper and the grid was dried at room temperature. The micrographs were acquired using an electron acceleration voltage of 200 kV. The elemental analysis was carried out using the EDX detector (X-MaxN 80T—Oxford Instruments, Abingdon, Oxfordshire, UK). The diameter of the nanoparticles was measured with ImageJ 1.53k and the histogram of the distribution was plotted with OriginPro 9.9.5, with a kernel smooth curve type. The aggregates were measured as one particle.

#### 4.4.11. Dynamic Light Scattering (DLS) and Zeta Potential Analysis

DLS analysis and Zeta potential measurement were performed using the AMERIGO™—Particle Size & Zeta potential Analyzer (Cordouan Technologies, Pessac, France), as per the manufacturer’s instructions, and the result analysis was assessed with the AmeriQ 3.2.3.0 software.

#### 4.4.12. Measurement of Mucin Binding Efficiency

Mucin binding efficiency was assessed by periodic acid–Schiff reaction (PAS) [92,93]. Schiff’s reagent was prepared by dissolving 0.55 g of fuchsin basic in 100 mL of double-distilled boiling water. After reaching approximately 50 °C, 11 mL of 1N HCl was added. After reaching room temperature, 1.11 g of potassium metabisulphite was added. After stirring, the solution was incubated at room temperature in the dark for 24 h. Subsequently, 0.55 g of activated charcoal was added and, after stirring for a few seconds, it was filtered through filter paper until the solution became pale yellow. It was then stored at 4 °C, protected from light. Se-HNF was mixed with a suspension of 0.1% mucin prepared in double-distilled water (*w*/*v*) using a hydrogel/mucin ratio = 90.

The mixture was incubated at 37 °C (Static Cooled incubator MIR-154, PHCbi, Kansas, MO, USA) for 1 h under shaking (Trayster IKA, Staufen im Breisgau, Germany) and then centrifuged at room temperature at 20,000 RCF for 1 h. Then, 1 mL of supernatant was mixed with 100 μL of acid periodic solution prepared by adding 10 µL of 50% periodic acid to 7 mL of 7% acetic acid. The mixture was incubated for 2 h at 37 °C under shaking. Subsequently, 100 μL of Schiff’s reagent was added. After 30 min of incubation at room temperature in the dark, the absorbance was read at λ = 555 nm using a microplate reader. The mucin binding efficiency was calculated by subtracting the free mucin concentration from the initial mucin concentration. The calibration curve was performed in the concentration range of 0–0.07% from a 0.1% mucin stock solution.

#### 4.4.13. Scanning Electron Microscopy (SEM)

TM4000Plus II tabletop electron microscope (Hitachi, Tokyo, Japan) was used for acquiring SEM micrographs of freeze-dried samples with the following parameters: 15 kV electron acceleration voltage, 1000× magnification, secondary electrons (SE) detector and low-charge mode (L). The setups were made in accordance with the methodology provided by the manufacturer.

#### 4.4.14. Contact Angle and Interfacial Tension Measurements

The contact angle (°) and interfacial tension (mN/m) of Se-HNF were evaluated using the OCA50 optical tensiometer (DataPhysics Instruments, Filderstadt, Germany). The contact angle was measured using the sessile drop mode on polar (glass) and non-polar (polystyrene) surfaces, and the interfacial tension was measured using the pendant drop mode. Both analyses were performed using the software SCA20 version 5.0.41.

#### 4.4.15. Rheological Analysis

The rheological behavior of Se-HNF, Mu and Se-HNF-Mu was studied using HR20 equipment from TA Instruments (New Castle, DE, USA) with parallel plate geometry at different oscillation and shear rates in up and down (reverse-R) ramps to determine the hysteresis loops and the thixotropy. A three-step procedure at constant temperature (25 °C) was used, starting in step one from low shear rates to high shear rates in frequency sweep (oscillatory) mode in the angular frequencies range ω = 0.1–100 rad/s (radians/s) and hysteresis mode (increasing—UP, then decreasing—R angular frequency ω) in order to determine the storage modulus G′ (Pa), the loss modulus G″ (Pa), the complex viscosity η* (Pa·s), the phase angle δ (°) and the stiffness k (N·m/rad). Where available, the crossover modulus G′ = G″ was determined by cubic spline interpolation. In step two, higher shear rates in flow sweep mode in the range 1–300 s^−1^ were used, as well as in up and down ramps. The third and final step, named axial, consisted of determining the axial adhesion force using a constant speed of 10 µm/s geometry lifting for 10 min. The adhesion energy (A_E_) was calculated as a work of adhesion with the formula: A_E_ = F × d/S, where F is the axial adhesion force, d is the work distance interpolated at the value of null axial force (or last negative value of F) and S is the contact surface with the 40 mm diameter geometry (cylinder plate), meaning S = 1.257 × 10^−3^ m^2^. The axial force curve was fitted with an exponential curve in which the *a* parameter defines the asymptote to the Y axis, the *b* parameter defines the asymptote to plateau and the *c* parameter is the power exponent, and it was interpreted as the speed of detachment of the hydrogel from the parallel plates. The viscosity and stress curves in flow sweep were fitted with the available functions in the Trios software version 5.1.1 from TA Instruments, respectively, Cross, Carreau, Carreau–Yasuda, Sisko and Williamson models for viscosity and, respectively, Newtonian, Bingham, Casson, power-law (Ostwald de Waele) and Herschel–Bulkley models for stress.

### 4.5. Assessement of In Vitro Biocompatibility of SeNPsSb and Se-HNF

In order to determine cell viability and proliferation, the CCK-8 assay was combined with the LIVE/DEAD assay. HGF-1 cells were seeded into 48-well plates using a cell density of 1 × 10^4^ cells/cm^2^. The complete culture medium consisted of Dulbecco’s Minimum Essential Media (DMEM) supplemented with 10% fetal bovine serum (FBS). Plates were incubated at 37 °C under 5% CO_2_ atmosphere throughout the experiment. After 24 h, the cells were treated with suspensions containing different concentrations of freeze-dried SeNPsSb (0.1, 0.5, 2.5, 10 µg/mL) or hydrogel (10, 25, 50, 100, 500, 1000 µg/mL) enriched with two concentrations of SeNPsSb (0.5 Se-HNF, 2.5 Se-HNF) and without SeNPsSb (HNF) in the complete culture medium.

For comparison, samples were followed by the negative cytotoxicity control (C−, untreated cells) and the positive cytotoxicity control (C+, cells treated with 7.5% dimethyl sulfoxide—DMSO). After 24 h, the suspensions were replaced with CCK-8 reagent, and the number of metabolically active cells was quantified spectrophotometrically using a microplate reader [27].

The gingival fibroblasts treated with the aforementioned suspensions were washed with serum-free culture medium, after which the LIVE/DEAD reagent was prepared and added according to [27]. After 10 min of incubation, the reagent was removed and the cells were washed again with serum-free culture medium. Subsequently, cells were visualized with the CelenaX High Content Imaging System (Logos Biosystems, Annandale, VA, USA). Image acquisition was conducted using CelenaX Explorer software version 1.0.5 (4× objective).

Cell morphology was revealed by fluorescent labelling of actin filaments with Alexa Fluor 488-coupled phalloidin, and the nuclei were stained with DAPI (4′,6-diamidino-2-phenylindole) [27]. The fluorescence microscopy images were acquired with CelenaX Explorer software (10× objective).

### 4.6. Determination of In Vitro Antioxidant Activity of SeNPsSb and Se-HNF

In order to determine the in vitro antioxidant activity of the products, the cells were seeded in 48-well plates using a cell density of 1 × 10^4^ cells/cm^2^. Throughout the experiment, the gingival fibroblasts were maintained at 37 °C under 5% CO_2_ atmosphere. At 24 h after seeding, the cells were treated with different concentrations of freeze-dried SeNPsSb (0.5, 2.5 µg/mL) or hydrogel (25, 1000 µg/mL) enriched with two concentrations of SeNPsSb (0.5 Se-HNF, 2.5 Se-HNF) and without SeNPsSb (HNF) in the presence of reactive oxygen species (ROS) inducer, i.e., a 37 µM hydrogen peroxide solution. All the suspensions were prepared in complete culture medium.

For comparison, samples were followed by the negative control (C−, untreated cells) and the positive control (C+, cells treated with 37 µM H_2_O_2_). At 24 h after treatment, the gingival fibroblasts were incubated with 2′,7′-dichlorodihydrofluorescein diacetate (H_2_DCFDA) solution for the labelling and quantifying of total reactive oxygen species (ROS) [94]. The fluorescence microscopy images were acquired with Celena X Explorer software (4× objective). For quantitative analysis, the fluorescence intensity was measured using a microplate reader (485 nm excitation, 530 nm emission) [94].

### 4.7. Investigation of Antimicrobial and Antibiofilm Activity of SeNPsSb, and Se-HNF

The diffusimetric method was used for the semiquantitative screening of the antimicrobial activity by droplet distribution of 10 µL of SeNPsSb prepared in double-distilled water and 25 µL of Se-HNF onto Müeller–Hinton agar (MHA) and Sabouraud dextrose agar (SDA), previously seeded with standardized microbial suspensions of 0.5 McFarland prepared in sterile physiological water from fresh 24 h cultures (MHA for *B. cereus*, *E. faecalis*a and *S. aureus* and SDA for *C. albicans*). After seeding, the Petri dishes were incubated at 37 °C with the lid down for 24 h [16]. The microbicidal effect of the tested product was quantified by the absence of microbial growth around the spot using the ImageJ 1.53k software [95].

Quantitative screening of the antimicrobial activity of SeNPsSb was performed by the serial microdilution method (10 serial dilutions of the product) in 180 μL of Müeller–Hinton medium in 96-well plates. The wells were seeded with 20 μL of microbial suspension of 0.5 McFarland cell density prepared in sterile physiological saline solution from fresh 24 h cultures. A sterility control (negative control) and a microbial culture control (positive control) were provided in each test. After incubation of the plates at 37 °C for 24 h, the absorbance was measured at 600 nm using a microplate reader. The lowest concentration of the product that inhibited the microbial growth represented the MIC value (mg/mL) of the product [96]. The SeNPsSb at the tested concentrations were also analyzed as a color control (without the microbial strain) using 20 µL of physiological saline solution, and the resulting optical densities were subtracted from the optical densities of the samples containing the microbial strain. The IC50 value (mg/mL) was determined as described in [97].

After quantitative screening of the antimicrobial activity of SeNPsSb, the wells were washed three times with 0.85% physiological saline solution. The bacterial biofilm was fixed with methanol for 5 min and stained with 0.1% crystal violet solution for 15 min. After washing the wells with 0.85% physiological saline solution, the stained bacterial biofilm was resuspended in 33% acetic acid, and the absorbance was measured at 490 nm using a microplate reader.

For the quantitative screening of the antimicrobial activity of Se-HNF at different cell densities, we chose a dose that was found to be biocompatible after performing the CCK-8 assay and that also exhibited an increased antioxidant activity. Therefore, a 25 µg/mL hydrogel enriched with two concentrations of SeNPsSb, i.e., 0.5 and 2.5 µg/mL (0.5 Se-HNF, 2.5 Se-HNF), and without (HNF) was added in MHB for *B. cereus*, *E. faecalis* and *S. aureus* and in SDB for *C. albicans.* Afterwards, 180 µL of each hydrogel suspension was transferred in 96-well plates. Subsequently, 20 µL of serial dilutions of microbial suspension (1.5 × 10^8^, 1.5 × 10^7^, 1.5 × 10^6^, 1.5 × 10^5^, 1.5 × 10^4^, 1.5 × 10^3^, 1.5 × 10^2^, 1.5 × 10^1^) prepared in 0.85% physiological saline solution were added to the previous suspensions. Optical density values were measured at 600 nm using a microplate reader. Sterility control (C−, negative control) and a microbial culture control (C+, positive control) for each microbial density were provided for each test [16].

## 5. Conclusions

Selenium nanoparticles were phytosynthesized with sea buckthorn leaf extract (SeNPsSb) and embedded in a hydrogel nanoformulation (Se-HNF) based on never-dried bacterial nanocellulose (NDBNC) and water-soluble chitosan (CS). The FTIR analysis suggested a hydrogen-bonded biocorona of various biocompounds for SeNPsSb and multiple molecular interactions between the hydrogel components. XRD-WAXS and SAXS suggest that the SeNPs nanograins are clusters of multiple 3 nm crystallites, whereas the amorphous character may be related to the biocorona. XRD-SAXS analysis evidenced a polydisperse distribution of SeNPsSb concentrated between 9 and 60 nm which was confirmed by TEM image analysis, whereas DLS analysis evidenced hydrated nano- and microparticles between 4 and 650 nm. SeNPsSb and Se-HNF exhibited a high degree of cytocompatibility, as well as antioxidant, antimicrobial and antibiofilm performance. The mucin binding efficiency assay, SEM analysis, as well as the rheological studies, supported the mucoadhesive potential of the SE-HNF. The adhesion force of Se-HNF was 0.186 N, the maximum adhesion time was 34 s and the calculated adhesion energy was 0.3 J/m^2^, while after the contact with diluted mucin suspension, the adhesion force decreased to 0.08 N, the adhesion time decreased to 5 s and the adhesion energy decreased to 0.08 J/m^2^, suggesting a high binding efficiency estimated to be around 72%. Altogether, the Se-HNF hydrogel nanoformulation has the potential to disperse the dysbiotic bacterial biofilm and to re-establish the homeostasis of the oral cavity.

## Figures and Tables

**Figure 1 pharmaceuticals-17-00023-f001:**
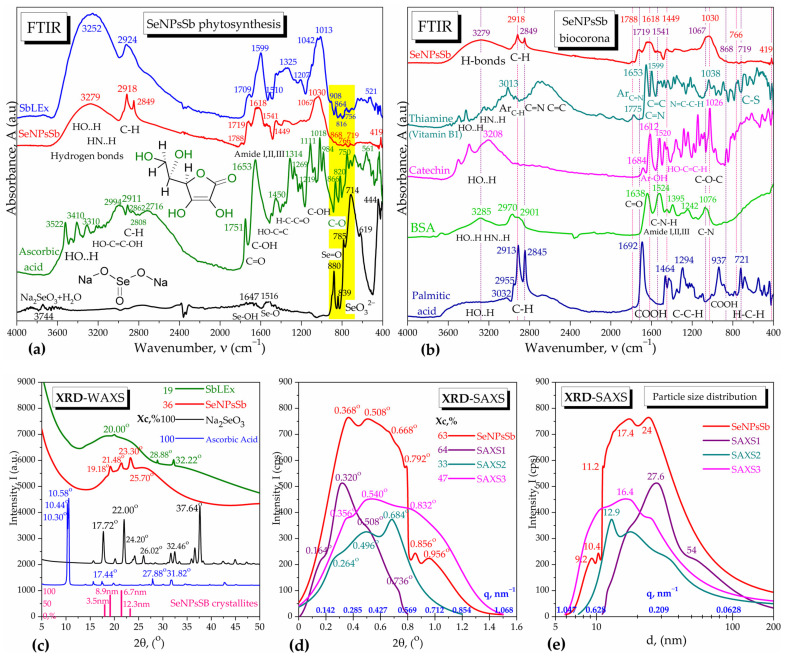
ATR-FTIR spectra and XRD analyses of the selenium nanoparticles (SeNPsSb): (**a**,**b**) ATR-FTIR spectra of SeNPsSb compared with: (**a**) Na_2_SeO_3_, ascorbic acid and sea buckthorn leaves aqueous extract (SbLEx); (**b**) thiamine, catechin, BSA and palmitic acid; (**c**) XRD-WAXS of SeNPsSb compared with the extract SbLEx, Na_2_SeO_3_ and ascorbic acid; (**d**) XRD-SAXS of SeNPsSb presented as a convoluted signal between three different analyses, with SAXS1, SAXS2 and SAXS3 performed between the 2θ angles 0–0.8°, 0–1.2° and 0–1.5°, respectively; (**e**) particle size distribution of SeNPsSb determined by XRD-SAXS.

**Figure 2 pharmaceuticals-17-00023-f002:**
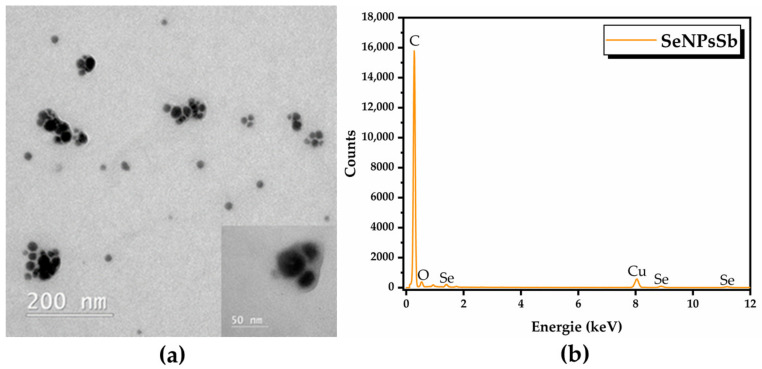
TEM-EDX analysis of the selenium nanoparticles (SeNPsSb): (**a**) TEM analysis; (**b**) EDX analysis.

**Figure 3 pharmaceuticals-17-00023-f003:**
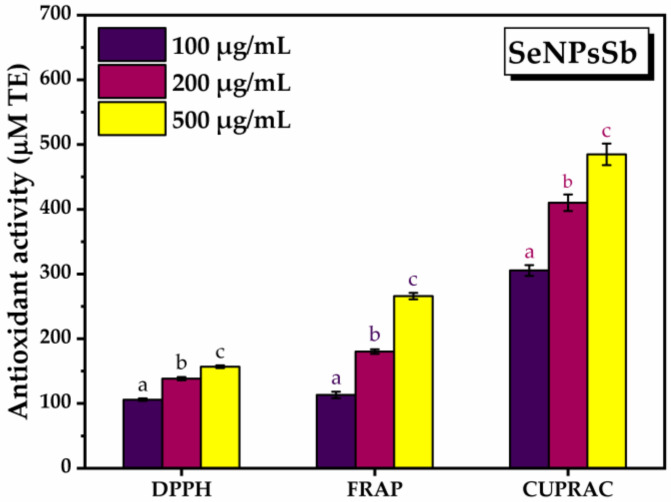
Antioxidant activity of the selenium nanoparticles (SeNPsSb) at different concentrations (±error bars, α < 0.05, *n* = 3; different letters mean statistically significant differences between the samples); DPPH: 2,2-diphenyl-1-picrylhydrazyl, FRAP: ferric reducing antioxidant power, CUPRAC: cupric reducing antioxidant activity.

**Figure 4 pharmaceuticals-17-00023-f004:**
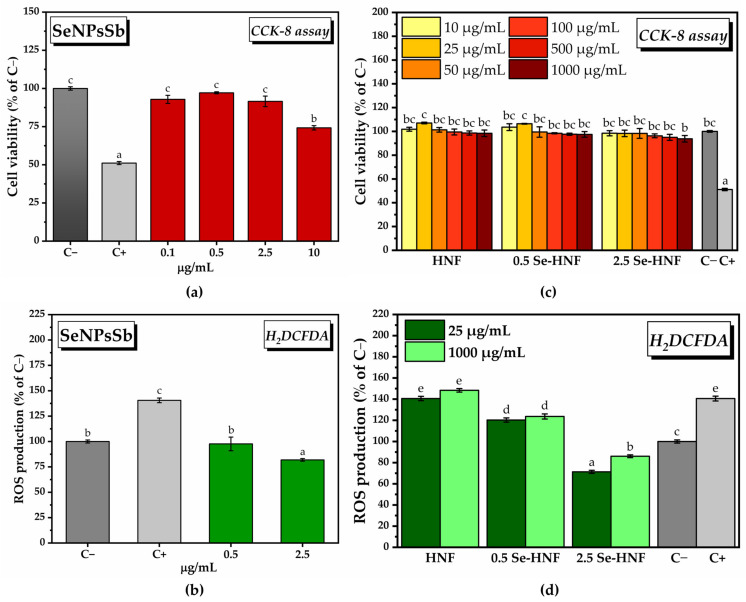
Cytocompatibility and in vitro antioxidant activity of SeNPsSb, HNF and Se-HNF: (**a**) CCK-8 assay for SeNPsSb; C− (untreated cells, negative cytotoxicity control), C+ (cells treated with 7.5% dimethyl sulfoxide (DMSO), positive cytotoxicity control); (**b**) quantifying total ROS after H_2_DCFDA labeling of HGF-1 cells treated with SeNPsSb; C− (untreated cells, negative control), C+ (cells treated with 37 µM H_2_O_2_-ROS inducer, positive control); (**c**) CCK-8 assay for Se-HNF; C− (untreated cells, negative cytotoxicity control), C+ (cells treated with 7.5% dimethyl sulfoxide (DMSO), positive cytotoxicity control), HNF—5% water-soluble chitosan in 0.4% never-dried bacterial nanocellulose; 0.5 Se-HNF—HNF with 0.5 µg/mL SeNPsSb; 2.5 Se-HNF—HNF with 2.5 µg/mL SeNPsSb; (**d**) quantifying total ROS after H_2_DCFDA labeling of HGF-1 cells treated with Se-HNF; C− (untreated cells, negative control), C+ (cells treated with 37 µM H_2_O_2_-ROS inducer, positive control), HNF—5% water-soluble chitosan in 0.4% never-dried bacterial nanocellulose; 0.5 Se-HNF—HNF with 0.5 µg/mL SeNPsSb; 2.5 Se-HNF—HNF with 2.5 µg/mL SeNPsSb (±error bars, α < 0.05, *n* = 3; different letters mean statistically significant differences between the samples).

**Figure 5 pharmaceuticals-17-00023-f005:**
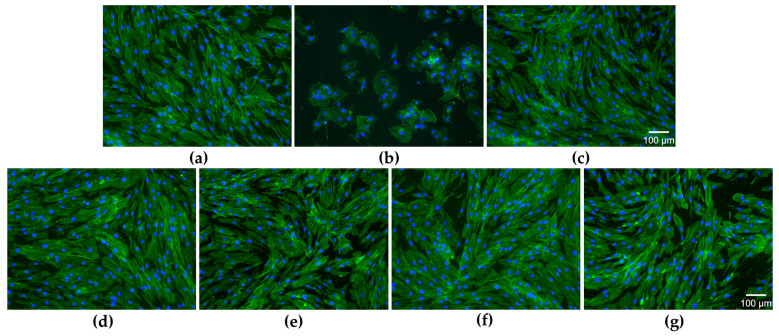
Effect of SeNPsSb, HNF and Se-HNF treatment on the HGF-1 cell morphology: (**a**) untreated cells (negative cytotoxicity control); (**b**) cells treated with 7.5% DMSO (positive cytotoxicity control); (**c**) 2.5 µg/mL SeNPsSb; (**d**) 25 µg/mL HNF; (**e**) 1000 µg/mL HNF; (**f**) 25 µg/mL 2.5 Se-HNF; (**g**) 1000 µg/mL 2.5 Se-HNF; HNF—5% water-soluble chitosan in 0.4% never-dried bacterial nanocellulose; 2.5 Se-HNF—HNF with 2.5 µg/mL SeNPsSb (green fluorescence indicates the labeled actin cytoskeleton and blue fluorescence indicates the stained nuclei).

**Figure 6 pharmaceuticals-17-00023-f006:**
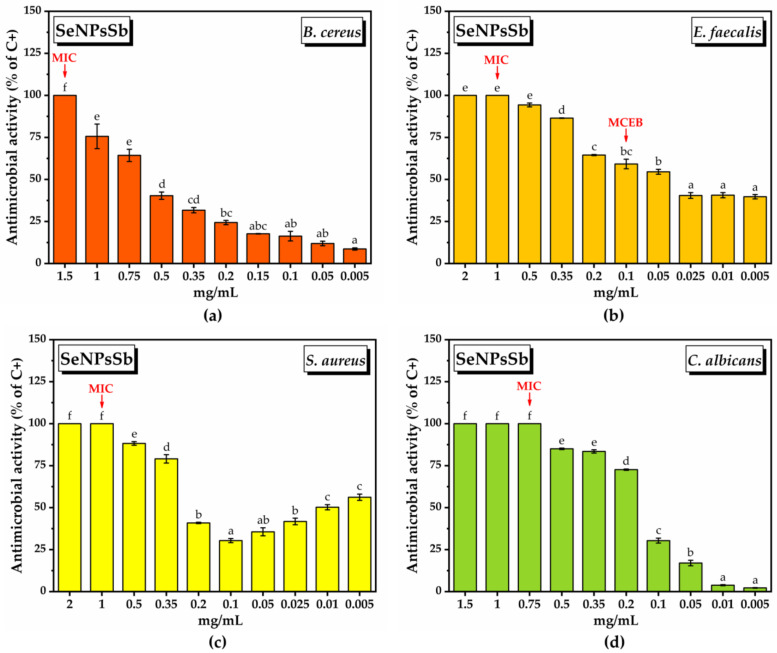
Antimicrobial activity of SeNPsSb at 24 h after treatment: (**a**) inhibition of *B. cereus* growth; (**b**) inhibition of *E. faecalis* growth; (**c**) inhibition of *S. aureus* growth; (**d**) inhibition of *C. albicans* growth (±error bars, α < 0.05, *n* = 3; different letters mean statistically significant differences between the samples).

**Figure 7 pharmaceuticals-17-00023-f007:**
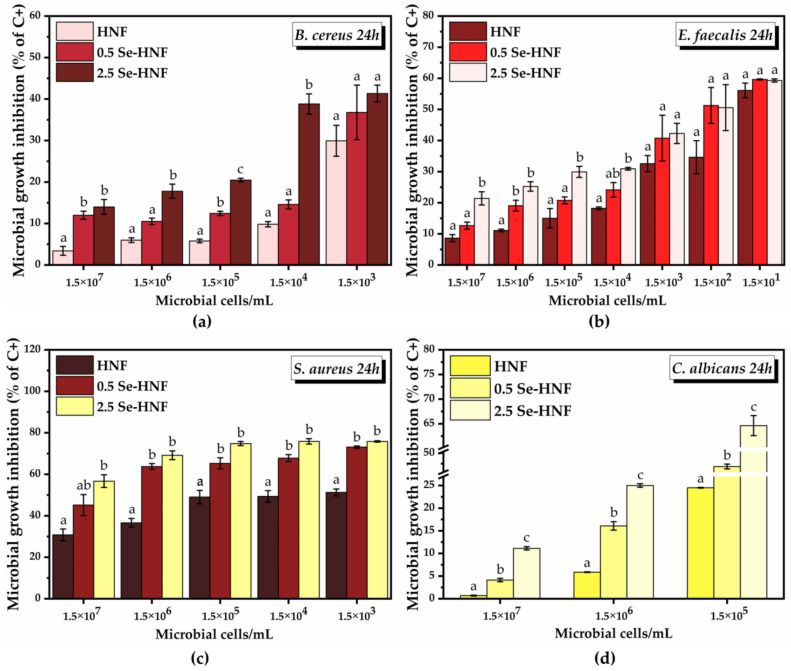
Quantitative screening of the antibacterial activity of hydrogel formulations: (**a**) *B. cereus* growth inhibition 24 h after hydrogel treatment; (**b**) *E. faecalis* growth inhibition 24 h after hydrogel treatment; (**c**) *S. aureus* growth inhibition 24 h after hydrogel treatment; (**d**) *C. albicans* growth inhibition 24 h after hydrogel treatment; (±error bars, α < 0.05, *n* = 3; different letters mean statistically significant differences between the samples; each microbial density was analyzed separately for statistical significance investigation); HNF—5% water-soluble chitosan in 0.4% never-dried bacterial nanocellulose; 0.5 Se-HNF—HNF with 0.5 µg/mL SeNPsSb; 2.5 Se-HNF—HNF with 2.5 µg/mL SeNPsSb.

**Figure 8 pharmaceuticals-17-00023-f008:**
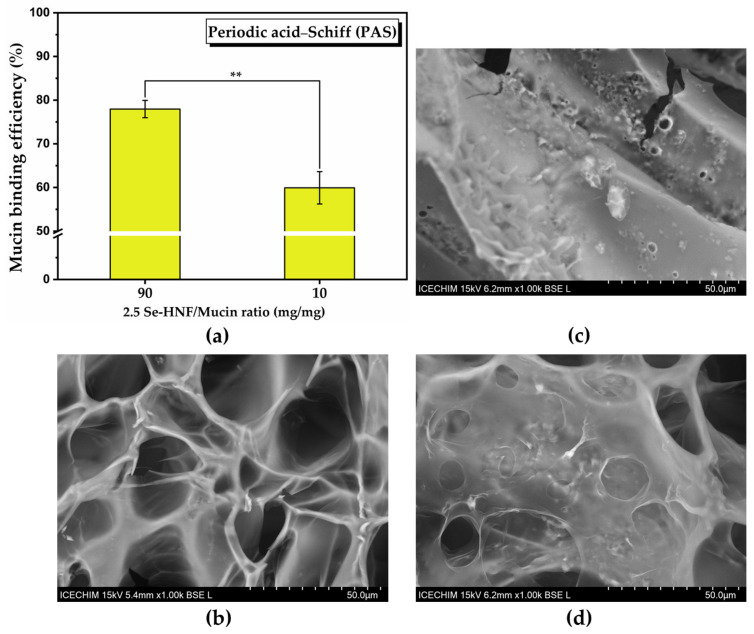
Mucin binding efficiency and SEM analysis of 2.5 Se-HNF: (**a**) periodic acid–Schiff (PAS) (±standard deviation, α < 0.5, *n* = 3, **—σ between 0.01 and 0.001); (**b**–**d**) SEM analyses of: (**b**) 2.5 Se-HNF; (**c**) mucin (Mu); (**d**) 2.5 Se-HNF-Mu; 2.5 Se-HNF—5% water-soluble chitosan in 0.4% never-dried bacterial nanocellulose enriched with 2.5 µg/mL SeNPsSb; Mu—3.5% aqueous mucin suspension; 2.5 Se-HNF-Mu—2.5 Se-HNF mixed with the 3.5% aqueous mucin suspension at the ratio 1:1 (*v*/*v*).

**Figure 9 pharmaceuticals-17-00023-f009:**
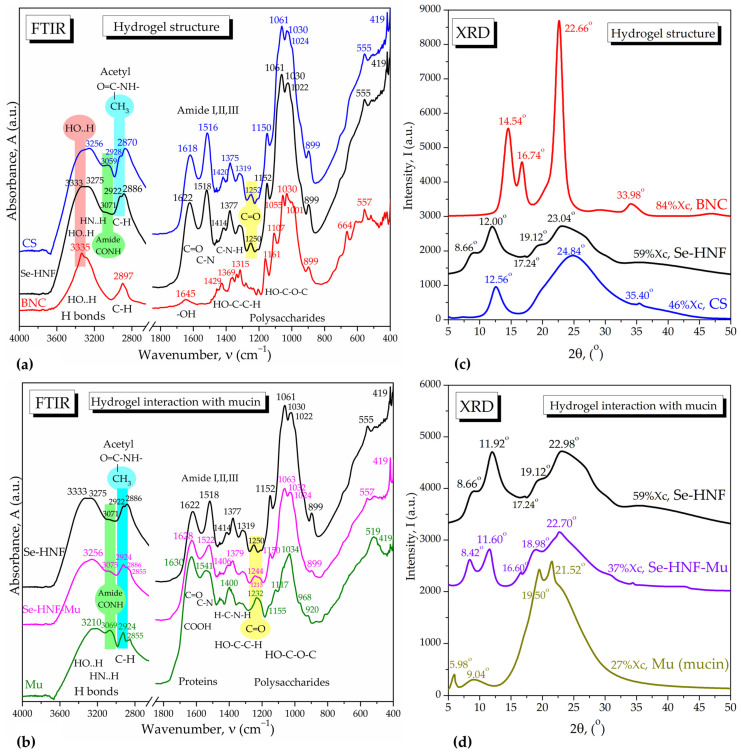
ATR-FTIR spectra and XRD structure analysis of the 2.5 Se-HNF hydrogel and its interaction with mucin: (**a**,**b**) ATR-FTIR spectra of 2.5 Se-HNF compared with: (**a**) chitosan (CS) and bacterial nanocellulose (BNC); (**b**) mucin (Mu) and 2.5 Se-HNF-Mu system; (**c**) 2.5 Se-HNF diffractogram compared with the diffractograms of bacterial nanocellulose (BNC) and chitosan (CS); (**d**) Se-HNF diffractogram compared with the diffractograms of mucin (Mu) and the Se-HNF-Mu system; 2.5 Se-HNF—5% water-soluble chitosan in 0.4% never-dried bacterial nanocellulose enriched with 2.5 µg/mL SeNPsSb; Mu—3.5% aqueous mucin suspension; 2.5 Se-HNF-Mu—2.5 Se-HNF mixed with the 3.5% aqueous mucin suspension at the ratio 1:1 (*v*/*v*).

**Figure 10 pharmaceuticals-17-00023-f010:**
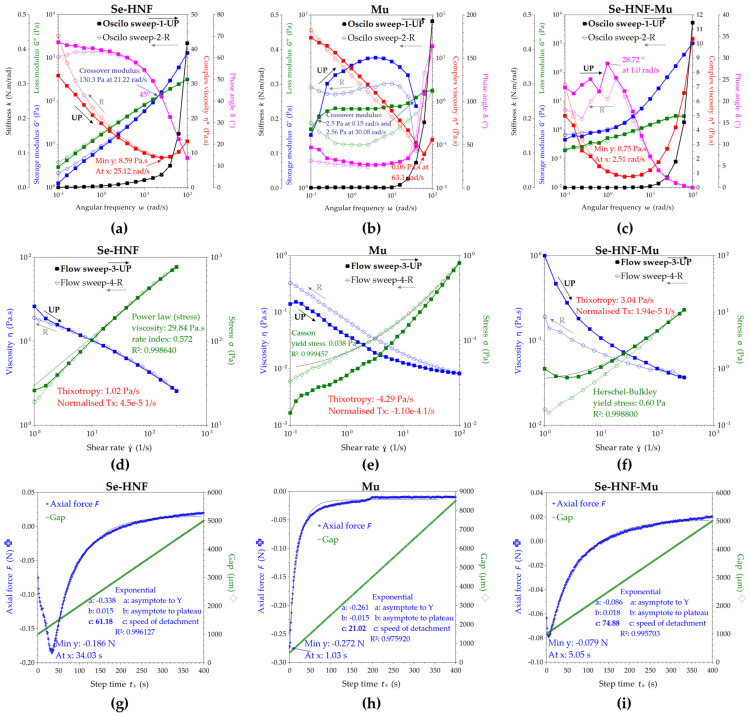
Rheological properties determined: (**a**) in oscillatory mode with hysteresis for the hydrogel 2.5 Se-HNF; (**b**) in oscillatory mode with hysteresis for Mu; (**c**) in oscillatory mode with hysteresis for 2.5 Se-HNF-Mu; (**d**) in hysteresis shear mode for hydrogel 2.5 Se-HNF; (**e**) in hysteresis shear mode for Mu; (**f**) in hysteresis shear mode for hydrogel with mucin 2.5 Se-HNF-Mu; (**g**) in axial mode for the hydrogel 2.5 Se-HNF; (**h**) in axial mode for Mu; (**i**) in axial mode for hydrogel with mucin 2.5 Se-HNF-Mu; 2.5 Se-HNF—5% water-soluble chitosan in 0.4% never-dried bacterial nanocellulose enriched with 2.5 µg/mL SeNPsSb; Mu—3.5% aqueous mucin suspension; 2.5 Se-HNF-Mu—2.5 Se-HNF mixed with the 3.5% aqueous mucin suspension at the ratio 1:1 (*v*/*v*).

**Table 4 pharmaceuticals-17-00023-t004:** Semiquantitative screening of antimicrobial activity of SeNPsSb.

Average Diameter of the Inhibition Zone (cm) ± Standard Error (SE)						
Concentration (mg/mL)	0.05	0.1	0.2	0.5	1	2
*B. cereus*	* R	* R	* R	0.84 ± 0.01 a **	0.88 ± 0.03 a	0.87 ± 0.01 a
*E. faecalis*	* R	0.39 ± 0.001 a	0.51 ± 0.006 b	0.86 ± 0.009 c	0.86 ± 0.02 c	0.906 ± 0.03 c
*S. aureus*	0.86 ± 0.01 a	0.86 ± 0.008 a	0.89 ± 0.03 a	0.98 ± 0.03 a	1.04 ± 0.07 a	1.02 ± 0.06 a
*C. albicans*	* R	* R	* R	* R	* R	* R

* Resistant. ** Different letters mean statistically significant differences between the samples.

**Table 5 pharmaceuticals-17-00023-t005:** MIC, IC50 and antibiofilm activity of SeNPsSb 24 h after treatment.

Strain	MIC (mg/mL)	IC50 (mg/mL) ± SE		MCEB (mg/mL)
*B. cereus*	1.5	0.708 ± 0.031		-
*E. faecalis*	1	0.039 ± 0.008		0.1
*S. aureus*	1	0.288 ± 0.004		-
*C. albicans*	0.75	0.143 ± 0.001		-
Bacterial biofilm inhibition ± standard error 24 h after SeNPsSb treatment				
*E. faecalis*/ at SeNPsSb (mg/mL)	81.22 ± 3.93/ 0.05	64.19 ± 2.62/ 0.025	57.64 ± 3.93/ 0.01	36.68 ± 1.31/ 0.005

**Table 7 pharmaceuticals-17-00023-t007:** Contact angle (°) and interfacial tension (mN/m) of 2.5 Se-HNF *.

Contact Angle (°) ± SD	Interfacial Tension (mN/m) ± SD
Glass	Polystyrene
52.82 ± 1.23	73.85 ± 0.39	97.6 ± 0.47
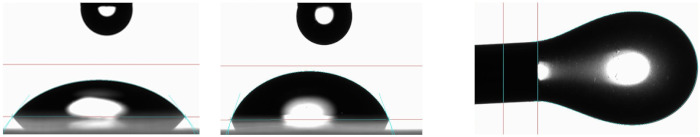

* 2.5 Se-HNF—5% water-soluble chitosan in 0.4% never-dried bacterial nanocellulose enriched with 2.5 µg/mL SeNPsSb.

## Data Availability

All the data are presented in this work.

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
