# Peer review of "Cytocompatibility, Antimicrobial and Antioxidant Activity of a Mucoadhesive Biopolymeric Hydrogel Embedding Selenium Nanoparticles Phytosynthesized by Sea Buckthorn Leaf Extract"

_pharmaceuticals, 2023, doi:10.3390/ph17010023_

Round 1

Reviewer 1 Report

Comments and Suggestions for Authors

The authors present a multi-functional biomaterial with thorough characterization. The manuscript is hindered by poor organization though. There is a lot of data presented here, which could even be enough for 2 separate manuscripts. More thought should be put into how the information is presented as the authors address the below comments.

1.       The abstract should include more information about the results vs. the methods used.

2.       Abbreviations should be defined before use, not after (e.g., MAMPs (‘Microbial Associated Molecular Patterns)’ should be re-written as: ‘Microbial Associated Molecular Patterns (MAMPs)’).

3.       The introduction does not flow well. In particular, paragraphs 2-5 read as completely separate pieces of information, with no rationale for how or why they are connected in this material system.

4.       The format of manuscripts in Pharmaceuticals includes Materials and Methods before Results. Please re-organized to fit this format.

5.       There are too many figures and tables. Please move less essential results to supplemental data and compile figures into 8 or fewer total figures in the main manuscript. For example, all of the data that shows successful synthesis could be compiled into a single figure with multiple parts. Cell images could be moved into the supplemental data, and all quantified cell data could be compiled into a single figure with multiple parts.

6.       The antioxidant activity data in Section 2.2 and Figure 4 should be moved to be presented with the other antioxidant activity data in Section 2.3.

7.       Section 2.4/Figure 9: Since biocompatibility encompasses both safety and efficacy for an intended application and requires animal testing, the authors should use a different term here (e.g., cytocompatibility) to more accurately represent the work done in this study (also in Section 4.1 and 4.5).

8.        The disconnect between initial antimicrobial characterization in Section 2.3 and later characterization in Section 2.4 is confusing. It would make more sense to present all mammalian cell data together in one section, and then discuss the overview of antimicrobial properties and the more detailed antimicrobial properties of select formulations in the following section.

9.       Comment 8 applies similarly to the physicochemical characterizations in Sections 2.2 and 2.5. These should be compiled into 1 section.

10.   The methods should be in the same order as the results (e.g., antioxidant properties should be moved to after FTIR, etc.). Methods can also be compiled more (e.g., all FTIR methods in 1 section).

Author Response

            The authors present a multi-functional biomaterial with thorough characterization. The manuscript is hindered by poor organization though. There is a lot of data presented here, which could even be enough for 2 separate manuscripts. More thought should be put into how the information is presented as the authors address the below comments.

      Point 1. The abstract should include more information about the results vs. the methods used.

      Response 1. Thank you. We included more information about the results.

Point 2. Abbreviations should be defined before use, not after (e.g., MAMPs (‘Microbial Associated Molecular Patterns)’ should be re-written as: ‘Microbial Associated Molecular Patterns (MAMPs)’).

Response 2. Thank you. We defined all the abbreviations before use.

Point 3. The introduction does not flow well. In particular, paragraphs 2-5 read as completely separate pieces of information, with no rationale for how or why they are connected in this material system.

Response 3. Thank you. We improved the Introduction section. We have moved the paragraph regarding nanocellulose after the paragraph 3, being also part of the nanostructures category.

Point 4.  The format of manuscripts in Pharmaceuticals includes Materials and Methods before Results. Please re-organized to fit this format.

Response 4. We have checked again the Template of Pharmaceuticals Journal and apparently it did not change, being with the Results and Discussion sections before the Materials and Methods section. If requested by the Editors of Pharmaceuticals, we can update the manuscript accordingly.

Point 5. There are too many figures and tables. Please move less essential results to supplemental data and compile figures into 8 or fewer total figures in the main manuscript. For example, all of the data that shows successful synthesis could be compiled into a single figure with multiple parts. Cell images could be moved into the supplemental data, and all quantified cell data could be compiled into a single figure with multiple parts.

Response 5. Thank you for your suggestion. We moved the fluorescence microscopy images acquired after performing the LIVE/DEAD assay and after labeling the HGF-1 cells with H2DCFDA in the supplementary materials (Figures S8-S11). FTIR and XRD results of SeNPsSb were compiled in one figure (Figure 1). We also compiled mucin binding efficiency and SEM analyses in one figure (Figure 8), and the FTIR and XRD results of Se-HNF in one figure (Figure 9). We reduced the number of figures from 17 to 10, trying to keep the flow logic of the paper as well.

Point 6. The antioxidant activity data in Section 2.2 and Figure 4 should be moved to be presented with the other antioxidant activity data in Section 2.3.

Response 6. We moved the antioxidant activity of SeNPsSb determined by DPPH, FRAP and CUPRAC assays in the section 2.3.

Point 7. Section 2.4/Figure 9: Since biocompatibility encompasses both safety and efficacy for an intended application and requires animal testing, the authors should use a different term here (e.g., cytocompatibility) to more accurately represent the work done in this study (also in Section 4.1 and 4.5).

Response 7. Thank you for your suggestion. In the places where we did not mention that it was in vitro biocompatibility, we changed into cytocompatibility, including in the title of the manuscript.

Point 8. The disconnect between initial antimicrobial characterization in Section 2.3 and later characterization in Section 2.4 is confusing. It would make more sense to present all mammalian cell data together in one section, and then discuss the overview of antimicrobial properties and the more detailed antimicrobial properties of select formulations in the following section.

Response 8. The HGF-1 cell viability and in vitro antioxidant activity results are now presented in one figure (now Figure 4), just after the antioxidant activity determined by the biochemical methods. Next, we present the analysis of cell morphology and we moved the data regarding the antimicrobial activity of SeNPsSb next to the antimicrobial activity of Se-HNF, as suggested.

Point 9. Comment 8 applies similarly to the physicochemical characterizations in Sections 2.2 and 2.5. These should be compiled into 1 section.

Response 9. Thank you for the suggestion, but we cannot merge the physicochemical characterization data from sections 2.2 and 2.5 because in section 2.5 the analyses are performed on the final hydrogel which contains 2.5 µg/mL SeNPsSb (2.5 Se-HNF), concentration that was found optimal after investigating the biological performance. Moreover, in section 2.2. we demonstrated the phyto-synthesis of SeNPsSb, which requires some characterization such as TEM and we analyzed its physicochemical properties, which we consider to be essential before investigating the biological performance, these NPs being reported for the first time, to the best of our knowledge.

Point 10. The methods should be in the same order as the results (e.g., antioxidant properties should be moved to after FTIR, etc.). Methods can also be compiled more (e.g., all FTIR methods in 1 section).

Response 10. We compiled the FTIR and XRD methods in one section and we changed the order of the materials and methods section according to your suggestion. Thank you very much for all your suggestions.

Reviewer 2 Report

Comments and Suggestions for Authors

Naomi Tritean et al… reports a good paper entitled as “Biocompatibility, Antimicrobial and Antioxidant Activity of a Mucoadhesive Biopolymeric Hydrogel Embedding Selenium Nanoparticles Phyto-synthesized by Sea Buckthorn Leaf Extract” This study aligns well with the journal's scope and is promising for future researchers. The theoretical and conceptual framework is solid; however, some clarifications are needed. 

Author Response

Naomi Tritean et al… reports a good paper entitled as “Biocompatibility, Antimicrobial and Antioxidant Activity of a Mucoadhesive Biopolymeric Hydrogel Embedding Selenium Nanoparticles Phyto-synthesized by Sea Buckthorn Leaf Extract” This study aligns well with the journal's scope and is promising for future researchers. The theoretical and conceptual framework is solid; however, some clarifications are needed. Upon addressing the following comments, the manuscript can be accepted for publication.

Point 1. Analysing the thermal properties of hydrogels and polymeric materials is essential. Are you conducting an examination of the thermal characteristics of hydrogel through Thermogravimetric Analysis (TGA)?

Response 1. Thank you very much for your appreciations and suggestions. Indeed, TGA would have been a relevant analytical method for the thermal stability and particular onset temperatures of the biopolymers, but the main purpose of this work was to check and optimize the biological activity of SeNPsSb and hydrogel and we tried to limit the study to the physico-chemical methods most correlated with this purpose, considering as well the specificity of the Journal Pharmaceuticals. Moreover, the paper would have become even longer than it already is. We plan to consider TGA as an additional method in future studies, in order to optimize further the hydrogel.

Point 2. In Figure 3, the EDX analysis reveals the presence of C, O, and Se as the main elements. Could you explain the relatively low intensity of the Se element peak? what are the weight and atomic percentages for each element?

Response 2. EDX analysis is mainly used to obtain qualitative information regarding the elemental composition of an analyzed sample surface. However, the analysis will almost always consist of secondary signals from the support used especially if the sample has nanometric thickness. In this case, a copper grid coated with a carbon film was used, which explains the high signal received for both carbon and copper. The purpose of this analysis was to confirm the selenium nature of the nanoparticles. However, from the spectrum presented in Figure 3 compositional information can be obtained. In this case, the composition is shown in the table below:

Element

Weight %

Atomic %

Uncert. %

C(K)

86.31%

92.89%

0.24

O(K)

7.22%

5.84%

0.15

Cu(K)

5.44%

1.10%

0.05

Se(K)

1.03%

0.17%

0.02

Total

100.00%

100.00%

            The spectrum shown in Figure 3 is obtained as a signal of the surface which is exposed to the electron beam to obtain the BF-TEM micrograph.

Point 3. What is the yield of SeNPs synthesized via photosynthesis using sea buckthorn leaves extract? Have you performed an evaluation?

Response 3. Thank you for your suggestion. We included the SeNPsSb yield at the beginning of the section 2.2. (lines 160-161), which was determined by the method described in section 4.2.1.

Point 4. You have mentioned that it is the chitosan and cellulose that form hydrogen bonds with the glycosidic groups of mucin in saliva which makes them good mucoadhesives. But what are the basic interactions that happen during the formation of Se-HNF between NDBNC, CS, and SeNPsSb?

Response 4. Thank you very much for these observations. Regarding SeNPsSb possible bonds and structure, we have added in the subsection 2.2., lines 166-169, the following text: “Hydrogen bonds could be formed between the non-bonding electrons of Se0 and hydrogen donors in biocorona [1, 2] and contribute together with van der Waals interactions to the stabilization of the NPs”. We also added before Fig. 1, at lines 187-188: " suggesting contributions from proteins and saccharides to the biocorona". Regarding the molecular interactions between the components of Se-HNF, we have added in the sub-section 2.5 before former Figure 15 (now Figure 9), lines 670-676: "The main interactions between NDBNC, CS and SeNPsSb within Se-HNF are the hydrogen bonds (H-bonds) between the proton-donors and functional groups containing non-bonding electrons, like oxygen in -OH, C=O, and COOH, and nitrogen in amines and amides. BNC and CS probably stabilize, additionally to the initial biocorona, the SeNPsSb via H-bonds and van der Waals interactions. These bonds and interactions induce band shifts in the FTIR spectrum of Se-HNF compared to the individual components, as visible in Figure 9a."

Point 5. Since it is claimed that the Se-HNF can be used to maintain oral cavity health by acting as a soft gel for removing dysbiotic biofilm, what is the pH compatibility of the formed hydrogel in the oral cavity?

Response 5. Thank you for your question. Since pH can be a biomarker of periodontal diseases, previous studies have indicated that in the case of healthy patients, the pH of the saliva is neutral, whereas the pH of the saliva of patients diagnosed with chronic gingivitis or chronic periodontitis can become slightly alkaline, respectively slightly acidic. Therefore, in the present study, we prepared a formulation with neutral pH to be able to support the physiological processes, so for this reason we used a water-soluble fungal chitosan which was solubilized in the nanocellulose suspension, previously prepared in double-distilled water. 

Point 6. Generally, hydrogels are hydrophilic substances crosslinked to hold water to a certain degree without being solubilized in water. In this work, as Se-HZN is intended for oralapplications leading to the direct contact of Se-HZN with saliva, which is predominantly composed of 99% water, what is the water absorption and retention capacity of Se-HZN?

Response 6. Thank you for your comment. We have added in the discussion section, last page: "behaves like a soft uncross-linked hydrogel with 95.18% water content that further dilutes in contact with saliva and gingival mucin, simulated as a 50% dilution with a 3.5% mucin aqueous solution, without losing its structural stability, as evidenced by rheology." Our hydrogel is not cross-linked and not intended for swelling and releasing of water, instead it interacts with saliva, gingival mucin and bacterial biofilm in order to disrupt the dysbiotic biofilm and re-establish the homeostasis of the oral cavity.

Point 7. In line 1083, the authors mention the MIC, but is there a comprehensive procedure outlined? Specifically, was any dye used in determining the MIC, and could you clarify this aspect?

Response 7. The serial microdilution method for MIC determination does not involve the use of a dye. It is based on the reading of the optical density at 600 nm. We also mentioned that because the selenium nanoparticle suspension is strongly colored (red-brown), the SeNPsSb at the tested concentrations were also analyzed as a color control (without the microbial strain) and the resulting optical densities were subtracted from the optical densities of the samples containing the microbial strain (lines 1388-1392). We also included a reference for the method used. We hope it is clearer now.

Point 8. Describe the procedure for preparing SeNPsSb and Se-HZN solutions for the experiment, specifying the solvent or medium used in this preparation process?

Response 8. Thank you. We included the solvent or the medium used in each assay.

Point 9. The authors conducted a live/dead cell assay, and Figure 6e indicates the presence of red coloration, signifying the observation of dead cells. Given the observed presence of dead cells, what prompted the authors to utilize a concentration of 2.5 µg/mL SeNPsSb for subsequent investigations?

Response 9. As mentioned, the CCK-8 assay can be correlated with the LIVE/DEAD assay because they both indicate cell viability, with the note that the CCK-8 assay is a quantitative one, while the LIVE/DEAD assay is only qualitative. The CCK-8 assay involves the reduction of the tetrazolium salt, WST-8, by cellular dehydrogenases to a yellow formazan, which is soluble in the culture medium and is spectrophotometrically evaluated at λ = 450 nm. Instead, the LIVE/DEAD assay contains two compounds which act simultaneously: calcein acetoxymethyl ester (AM) and ethidium homodimer (EthD-1). Calcein AM reacts with intracellular esterases of viable cells and emits a green fluorescence at λ=494/517 nm. Ethidium homodimer penetrates the nucleus of disrupted plasma membrane cells, intercalates between the nitrogenous bases of nucleic acids and emits red fluorescence at λ=528/617 nm.

            The cell viability determined by CCK-8 assay was 97.14 ± 0.66% for 0.5 µg/mL and 91.53 ± 3.47 µg/mL for 2.5 µg/mL SeNPsSb. Both percentages indicate high cytocompatibility as no significant changes were observed compared to the negative cytotoxicity control (C-, untreated cells). Thus, although viability slightly decreased and a few fluorescent red cells can be observed in the 2.5 µg/mL treated culture following the acquisition of fluorescence microscopy images after performing the LIVE/DEAD assay, this decrease in viability is not statistically significant, and both concentrations of SeNPsSb (0.5, 2.5 ug/mL) can be considered highly cytocompatible, especially as the percentages are > 90%.

Point 10. Background noise was observed in specific figures pertaining to in vitro cell line studies. Provide clarification

Response 10. Thank you for the question. It is not clear what specific figures you refer to, but if you refer to the fluorescence microscopy images acquired after labelling the total ROS with H2DCFDA, it is not a background noise. that the background noise would mean that the whole image would tend to be light green, including the background. In the case of the positive control (C+), i.e., cells treated with hydrogen peroxide (ROS inducer), the cells are bright fluorescent green, the reactive oxygen species being distributed throughout the cell. By maintaining the cells in the presence of the ROS inducer and different products, a decrease in ROS production can be observed, and the cells become diffusely stained in green. When the level of ROS further decreases, the image becomes darker as a result of the lack of a fluorescent signal. We hope that we managed to clarify this aspect.

Point 11. Check the typographical errors?

Response 11. Thank you. We re-checked the entire manuscript.

Reviewer 3 Report

Comments and Suggestions for Authors

The present manuscript entitled “Biocompatibility, Antimicrobial and Antioxidant Activity of a Mucoadhesive Biopolymeric Hydrogel Embedding Selenium 3 Nanoparticles Phyto-synthesized by Sea Buckthorn Leaf Extract” by Tritean et al., describes the development of selenium-enriched hydrogel nano-formulation (Se-HNF) based on Never-dried bacterial nanocellulose from Kombucha fermentation and fungal chitosan with embedded biogenic SeNPs phyto-synthesized by an aqueous extract of sea buckthorn leaves (SbLEx) – SeNPsSb. Furthermore, the total phenolic content and antioxidant activity of SbLEx were determined using LC-MS. The authors report an interesting work. The objective and justification of the work are clear. I was pleased to review your manuscript and also congratulate the authors for their good work and presentation of results particularly in Figures. The authors performed enough characterization techniques in the present study such as TEM-EDX, DLS, Zeta potential, FTIR, and XRD. Therefore, I recommend it for publication. However, some minor issues are detailed below which need to be addressed before its final acceptance in Pharmaceuticals.

I advise the authors to take the following points into account while revising their manuscript.

Comment 1: There are some typographical and grammatical errors in the manuscript text, so the authors need to correct them in the revised manuscript. Carefully check the scientific names in the manuscript must be in italics, also correct the subscript errors. 

Comment 2: The whole manuscript must be cross-checked thoroughly for English editing, grammatical, spelling mistakes, and syntax errors. So, I suggest the author's English language should be polished.

Comment 3: In the abstract, please note that the full name of the abbreviation should be provided for the first time it appears in the abstract, so provide the full form of LC-MS, TEM-EDX, SEM-EDX, DLS, FTIR, XRD, DPPH, FRAP, and CUPRAC.

Comment 4: Include the particle size distribution of Figure 3 (a) using Image J software.

Comment 5: The authors mentioned SEM-EDX analysis in the abstract section. However, they provided only SEM images in Figure 14, so, add EDX analysis of SEM images.

Comment 6: In XRD analysis, I suggest the authors include the discussion of major peak positions with the supported literature and XRD Database.

Comment 7: Include the conclusion section with clear quantitative findings and more emphasis on the findings and their implications may be mentioned in the conclusion section.

Comment 8: The homogeneity of the reference section needs to be maintained. In some references, scientific names are written in regular form, not in italics (E.g. reference no. 36). So please check and revise according to the journal's instructions.

Comments on the Quality of English Language

Minor editing of English language required.

Author Response

The present manuscript entitled “Biocompatibility, Antimicrobial and Antioxidant Activity of a Mucoadhesive Biopolymeric Hydrogel Embedding Selenium 3 Nanoparticles Phyto-synthesized by Sea Buckthorn Leaf Extract” by Tritean et al., describes the development of selenium-enriched hydrogel nano-formulation (Se-HNF) based on Never-dried bacterial nanocellulose from Kombucha fermentation and fungal chitosan with embedded biogenic SeNPs phyto-synthesized by an aqueous extract of sea buckthorn leaves (SbLEx) – SeNPsSb. Furthermore, the total phenolic content and antioxidant activity of SbLEx were determined using LC-MS. The authors report an interesting work. The objective and justification of the work are clear. I was pleased to review your manuscript and also congratulate the authors for their good work and presentation of results particularly in Figures. The authors performed enough characterization techniques in the present study such as TEM-EDX, DLS, Zeta potential, FTIR, and XRD. Therefore, I recommend it for publication. However, some minor issues are detailed below which need to be addressed before its final acceptance in Pharmaceuticals.

          I advise the authors to take the following points into account while revising their manuscript.

Point 1. There are some typographical and grammatical errors in the manuscript text, so the authors need to correct them in the revised manuscript. Carefully check the scientific names in the manuscript must be in italics, also correct the subscript errors. 

Response 1. Thank you. We re-checked the entire manuscript. We corrected the subscript errors and the scientific names which should have been in italics.

Point 2. The whole manuscript must be cross-checked thoroughly for English editing, grammatical, spelling mistakes, and syntax errors. So, I suggest the author's English language should be polished.

Response 2. Thank you. We re-checked the entire manuscript.

Point 3. In the abstract, please note that the full name of the abbreviation should be provided for the first time it appears in the abstract, so provide the full form of LC-MS, TEM-EDX, SEM-EDX, DLS, FTIR, XRD, DPPH, FRAP, and CUPRAC.

Response 3. Thank you. We provided the full name of the abbreviations.

Point 4. Include the particle size distribution of Figure 3 (a) using Image J software.

Response 4. Thank you for your suggestion. We included in Figure S4 the particle size distribution, although we did not have a significant number of nanoparticles for an optimal statistic analysis. According to your suggestion, we included also the particle size distribution determined by XRD-SAXS (Figure 1e).

Point 5. The authors mentioned SEM-EDX analysis in the abstract section. However, they provided only SEM images in Figure 14, so, add EDX analysis of SEM images.

Response 5. Thank you very much for your observation. The physicochemical analyses were performed on the hydrogel enriched with the concentration of 2.5 µg/mL SeNPsSb which was optimal after investigating the biological performance, therefore EDX analysis was not suitable for the identification of such a low concentration of Se. We removed the EDX from the abstract section.

Point 6. In XRD analysis, I suggest the authors include the discussion of major peak positions with the supported literature and XRD Database.

Response 6. Thank you for recommendation, we have added in sub-section 2.5 the available PDXL codes and main diffraction peaks for chitosan, cellulose Iα, Iβ and amorphous cellulose in the XRD database and relevant references (lines 700-711).

Point 7. Include the conclusion section with clear quantitative findings and more emphasis on the findings and their implications may be mentioned in the conclusion section.

Response 7. We revised the Conclusions, thank you for suggestion.

Point 8. The homogeneity of the reference section needs to be maintained. In some references, scientific names are written in regular form, not in italics (E.g. reference no. 36). So please check and revise according to the journal's instructions.

Response 8. Thank you. We re-checked and revised. 

Round 2

Reviewer 1 Report

Comments and Suggestions for Authors

The authors made substantial improvements to the manuscript. The following concerns are remaining prior to this manuscript being ready for publication:

1.       The discussion should be expanded to help connect the concentrations of compounds required for effective antioxidant and antimicrobial activities vs. the cytocompatible concentrations.

2.       A clear delineation should be made in the results and discussion between testing compounds in solution vs. in hydrogels. There are a lot of acronyms that are hard to follow, so the authors should clearly state what they are characterizing in section headings and/or the body of the text. A transition point in which the authors state something to the effect of: “Based on X, Y, and Z data, we selected A compounds for introduction into hydrogels for further characterization.” Alternatively (or in addition), the authors could consistently add the word ‘hydrogels’ to any acronyms that are related to gels in the text and in section headings.

3.       There are still a lot of figures and tables. Any further combinations of figures and/or tables could help to streamline reading the manuscript.

Comments on the Quality of English Language

The English language has been improved with revisions, but could be improved further. 

Author Response

Response to Reviewer Comments

The authors made substantial improvements to the manuscript. The following concerns are remaining prior to this manuscript being ready for publication:

Point 1. The discussion should be expanded to help connect the concentrations of compounds required for effective antioxidant and antimicrobial activities vs. the cytocompatible concentrations.

  Response 1. Thank you for your suggestion. The information added in the Discussion section is highlighted in yellow. (lines 994-1006).

Point 2. A clear delineation should be made in the results and discussion between testing compounds in solution vs. in hydrogels. There are a lot of acronyms that are hard to follow, so the authors should clearly state what they are characterizing in section headings and/or the body of the text. A transition point in which the authors state something to the effect of: “Based on X, Y, and Z data, we selected A compounds for introduction into hydrogels for further characterization.” Alternatively (or in addition), the authors could consistently add the word ‘hydrogels’ to any acronyms that are related to gels in the text and in section headings.

Response 2. In the Section 2.3. Biological performance of SeNPsSb and Se-HNF before determining the cytocompatibility of 0.5 and 2.5 µg/mL SeNPsSb-enriched hydrogels (0.5 Se-HNF, 2.5 Se-HNF), we have included some information in the following statement: “To investigate the cytocompatibility of the hydrogels based on the BNC-CS matrix enriched with 0.5 µg/mL and 2.5 µg/mL SeNPsSb (0.5 Se-HNF and 2.5 Se-HNF, respectively), we tested multiple hydrogel concentrations (10, 25, 50, 100, 500, 1000 µg/mL) prepared in complete culture medium (lines 407-408). The BNC-CS matrix without SeNPsSb (HNF) was also tested in order to determine if there is a potential complementarity.” We have added the fact that the mentioned concentrations were prepared in complete culture medium.

We also added “of the hydrogel with SeNPsSb” (lines 298-299), before Se-HNF, to be easier to follow.

Also, prior to all the tests that were performed on the mucoadhesive hydrogel namely the qualitative screening of antimicrobial activity (Table 7) and physicochemical properties in Section 2.4, we added the following mention: Due to the fact that the concentration of 2.5 µg/mL SeNPsSb was the highest tested concentration that proved to be cytocompatible and also showed the highest potential to reduce both ROS production and microbial growth by assessing the biological performance of Se-HNF in solution (Figure 4 and Figure 7), we further characterized more in-depth the 2.5 Se-HNF hydrogel”. Moreover, in all the legends related to the undiluted hydrogel analysis, there is the following mention: “2.5 Se-HNF – 5% water-soluble chitosan in 0.4% never-dried bacterial nanocellulose enriched with 2.5 µg/mL SeNPsSb”. We completed also other legends with the products tested. Thank you and we hope that we have managed to clarify this aspect.

Point 3. There are still a lot of figures and tables. Any further combinations of figures and/or tables could help to streamline reading the manuscript.

Response 3. We moved the results from the Table 3, respectively the quantification of catechin (C), epicatechin (EC), quercetin 3-rutinoside (Que-rut) in SbLEx by HPLC analysis in the Table 2 in which LC-MS analysis of SbLEx is also presented, therefore Table 3 has been deleted and the rest of the tables have been renumbered. We also combined the former Table 7 "Bacterial biofilm inhibition 24 h after the SeNPsSb treatment" with now Table 5 “MIC, IC50, and antibiofilm activity of SeNPsSb, at 24h after the treatment”. We hope that we have managed to reduce sufficiently their number. Thank you again for all your meaningful suggestions.